# An ancient role for CYP73 monooxygenases in phenylpropanoid biosynthesis and embryophyte development

Samuel Knosp [ID][1], Lucie Kriegshauser[1,6], Kanade Tatsumi[1,7], Ludivine Malherbe[1], Mathieu Erhardt [ID][1], Gertrud Wiedemann [ID][2,8], Bénédicte Bakan[3], Takayuki Kohchi [ID][4], Ralf Reski [ID][2,5] & Hugues Renault [ID][1✉]

## Abstract

The phenylpropanoid pathway is one of the plant metabolic pathways most prominently linked to the transition to terrestrial life, but its evolution and early functions remain elusive. Here, we show that activity of the *t*-cinnamic acid 4-hydroxylase (C4H), the first plant-specific step in the pathway, emerged concomitantly with the *CYP73* gene family in a common ancestor of embryophytes. Through structural studies, we identify conserved CYP73 residues, including a crucial arginine, that have supported C4H activity since the early stages of its evolution. We further demonstrate that impairing C4H function *via CYP73* gene inactivation or inhibitor treatment in three bryophyte species—the moss *Physcomitrium patens*, the liverwort *Marchantia polymorpha* and the hornwort *Anthoceros agrestis*—consistently resulted in a shortage of phenylpropanoids and abnormal plant development. The latter could be rescued in the moss by exogenous supply of *p*-coumaric acid, the product of C4H. Our findings establish the emergence of the *CYP73* gene family as a foundational event in the development of the plant phenylpropanoid pathway, and underscore the deep-rooted function of the C4H enzyme in embryophyte biology.

**Keywords** Plant Evolution; Bryophytes; Cytochromes P450; Biopolymers; Cinnamic Acid 4-Hydroxylase
**Subject Categories** Evolution & Ecology; Plant Biology

## Introduction

Green plants (i.e., Viridiplantae) have evolved over approximately one billion years and have thrived wherever light was available thanks to their photosynthetic capabilities. The successful evolutionary history of Viridiplantae have yielded an astonishing diversity of forms—from unicellular marine algae to giant redwood trees—and propelled them as the most prevalent living group on Earth from a biomass standpoint (Bar-On et al, 2018).

Most of this biomass is found on land (Bar-On et al, 2018), reflecting the cornerstone functions performed by plants in terrestrial ecosystems, primarily as an entry point for solar energy. Recent phylogenetic evidences indicate that modern land plants, also known as embryophytes, have a unique origin and emerged from freshwater algae about half a billion years ago (Cheng et al, 2019; Morris et al, 2018). The radical change in habitat experienced by early embryophytes, an evolutionary milestone called terrestrialization, involved profound morpho-physiological adaptations. In particular, the ability to synthesize a wide range of metabolites was instrumental by mitigating effects of terrestrial constraints (e.g., UV, lack of buoyancy, drought) and by providing chemical mediators for fast-expanding ecological interactions.

One of the most iconic plant metabolisms associated with terrestrialization is the phenylpropanoid pathway, which generates a suite of phenolic compounds that effectively address terrestrial challenges (Vogt, 2010; Weng and Chapple, 2010). This pathway leads to the synthesis of widespread polyphenolic molecules (e.g., flavonoids) and phenolic esters/amides (e.g., chlorogenic acids), which act as powerful UV screens and antioxidants. The phenylpropanoid pathway also supplies precursors of the four hydrophobic polymers cutin, suberin, sporopollenin, and lignin that strengthen and waterproof the cell wall. These polymers form the framework of various apoplastic diffusion barriers (e.g., cuticle, pollen coat, Casparian strip) and thus help plants to shield their tissues from external aggressions, and to manage water and solutes efficiently. In accordance to these essential functions, hydrophobic biopolymers make up a significant fraction of embryophyte biomass. Lignin for instance is regarded as the second most abundant biopolymer on the planet after cellulose, accounting for *ca.* 30% of biosphere organic carbon (Boerjan et al, 2003).

In order to support high biomass production and multiple physiological functions, the phenylpropanoid pathway is meticulously organized and regulated (Mizutani et al, 1997; Bassard et al, 2012). The first three steps of this pathway, collectively known as the general phenylpropanoid pathway (Fig. 1A), are obligatory and hence process

[1]IBMP | Institut de biologie moléculaire des plantes, CNRS, University of Strasbourg, Strasbourg, France. [2]Plant Biotechnology, Faculty of Biology, University of Freiburg, 79104 Freiburg, Germany. [3]INRAE, Biopolymers, Interactions, Assemblies Research Unit, La Géraudière, Nantes, France. [4]Graduate School of Biostudies, Kyoto University, Kyoto, Japan. [5]Signalling Research Centres BIOSS and CIBSS, University of Freiburg, 79104 Freiburg, Germany. [6]Present address: Amatera Biosciences, Pépinière Genopole Entreprise, 91000 Evry, France. [7]Present address: Research Institute for Sustainable Humanosphere, Kyoto University, Kyoto, Japan. [8]Present address: Inselspital, University of Bern, 3010 Bern, Switzerland. ✉E-mail: hugues.renault@ibmp-cnrs.unistra.fr

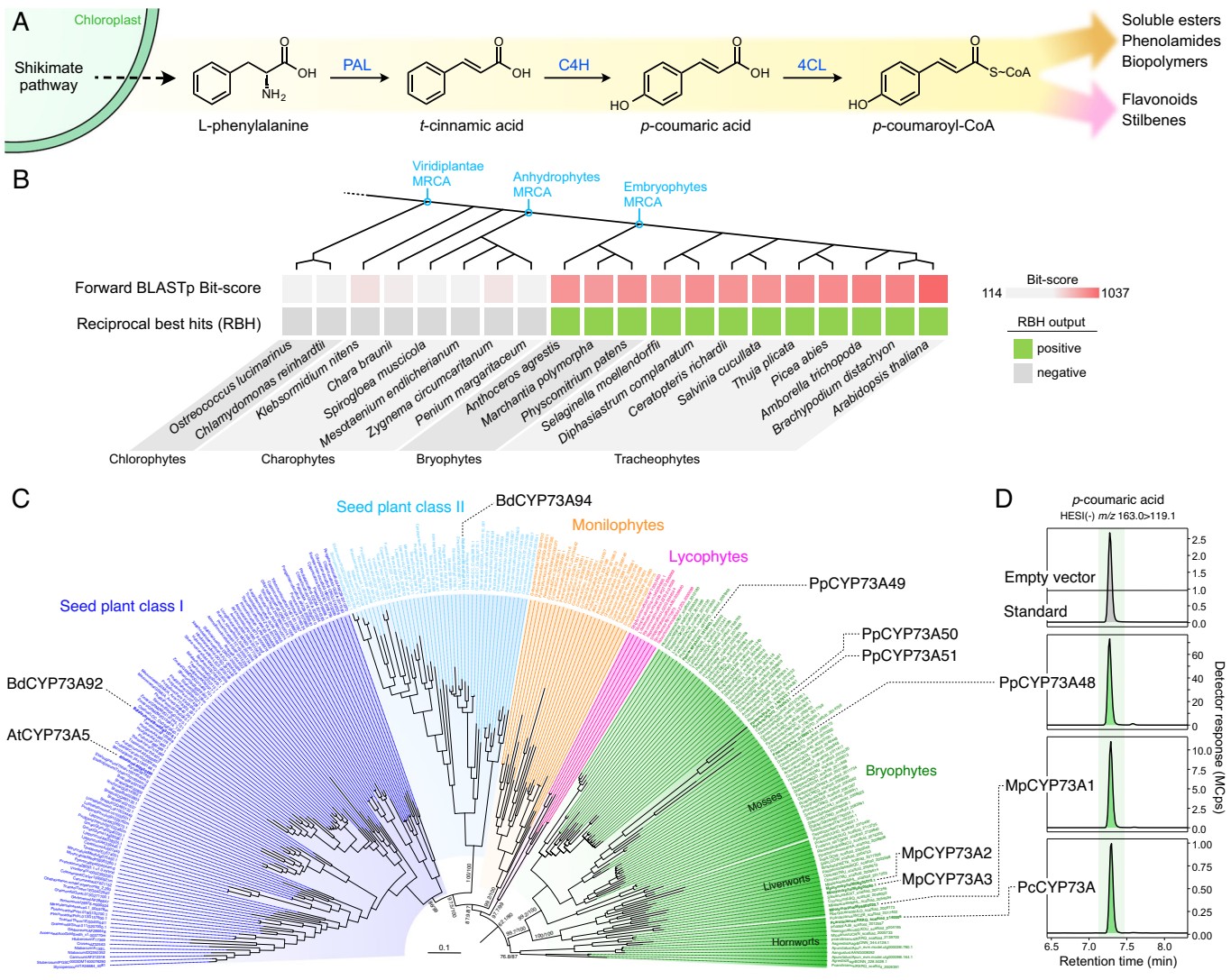

**Figure 1. Evolutionary history of the *CYP73* family encoding *t*-cinnamic acid 4-hydroxylase.**

(A) Schematic representation of the three steps of the general phenylpropanoid pathway. PAL, phenylalanine ammonia-lyase; C4H, cinnamic acid 4-hydroxylase; 4CL, 4-coumarate:CoA ligase. (B) Search for AtCYP73A5 homologs in Viridiplantae by a reciprocal best hit (RBH) strategy. MRCA, most recent common ancestor. (C) Maximum-likelihood nucleotide tree (IQ-TREE2, GTR + F + I + R7) describing the phylogenetic relationships between 275 *CYP73* homologous sequences. SH-aLRT test and ultrafast bootstrap (1000 pseudo-replicates) supports are annotated on main branches. CYP73 homologs relevant to the present study are indicated. At, *Arabidopsis thaliana*; Bd, *Brachypodium distachyon*; Mp, *Marchantia polymorpha*; Pc, *Phaeoceros carolinianus*. Scale bar represents the number of nucleotide substitutions per site. Tree was rooted according to the midpoint method. (D) Representative UHPLC-MS/MS chromatograms showing the production in vitro of *p*-coumaric acid from *t*-cinnamic acid (C4H activity) by recombinant CYP73 proteins representative of the three major bryophyte groups: mosses (PpCYP73A48), liverworts (MpCYP73A1) and hornworts (PcCYP73A). Assays performed with microsomes derived from yeasts transformed with an empty vector were used as negative controls. Source data are available online for this figure.

the entire metabolic flux. The initial step involves the deamination of phenylalanine by the phenylalanine ammonia-lyase (PAL) to produce *trans*-cinnamic acid, which is the first bona fide phenylpropanoid molecule. The phenolic ring of *t*-cinnamic acid is then hydroxylated at position 4 (*para* position) by the cinnamic acid 4-hydroxylase (C4H) to generate *para*-coumaric acid. Next, *p*-coumaric acid is activated with coenzyme A by 4-coumarate:CoA ligase (4CL), which leads to the formation of *p*-coumaroyl-CoA, a branch-point molecule that feeds into the various downstream pathways (Fig. 1A).

PAL activity is not exclusive to plants and is also found in cyanobacteria and fungi (Barros and Dixon, 2019), thus making C4H the first plant-specific step in the phenylpropanoid pathway (Renault et al, 2017b). C4H enzymes belong to the family 73 of cytochrome P450 monooxygenases (CYP73). Cytochromes P450 are highly diversified enzymes that catalyze irreversible and often rate-determining reactions and, as such, are considered to be key drivers of plant metabolism evolution and diversification, and control points in metabolic pathways (Hamberger and Bak, 2013; Liu et al, 2016;

Hansen et al, 2021). Accordingly, inactivation of the single *CYP73* gene from the tracheophyte model *Arabidopsis thaliana* led to dramatic developmental defects and phenylpropanoid deficiency (Schilmiller et al, 2009), pointing to the critical role of C4H in tracheophyte physiology. No such in planta investigation has been performed to date in a bryophyte model although these data appear key to improve our understanding of the evolution and early functions of C4H and, by extension, of the phenylpropanoid pathway as a whole.

Here we report a multidisciplinary study of C4H-encoding *CYP73* genes. We show that C4H activity emerged with the rise of the *CYP73* family in an embryophyte ancestor and identify key residues supporting its catalytic activity. We further show that impairing C4H function *via CYP73* gene inactivation or inhibitor treatment compromises phenylpropanoid biosynthesis and development in both tracheophyte and bryophyte species, pointing to a pivotal and conserved role of *CYP73* in embryophyte physiology.

## Results

### CYP73-catalyzed C4H activity originated in an embryophyte progenitor

Thus far, the only known enzymes capable of catalyzing the 4-hydroxylation of *t*-cinnamic acid (C4H activity) are the cytochrome P450 monooxygenases from the 73 family (CYP73). To investigate the evolution of C4H, a survey of *CYP73* genes was conducted using a reciprocal best hits (RBH) approach in 20 Viridiplantae genomes, including recent charophyte genomes. The results revealed that potential *CYP73* homologs were only found in bryophytes and tracheophytes, indicating that the origin of this CYP family can be traced back to a progenitor of embryophytes (Fig. 1B, Appendix Tab. S1). To bring further support to this evolutionary scenario, we performed a phylogenetic analysis of 275 *CYP73* sequences. We included 271 additional sequences from *A. thaliana*, *P. patens,* and charophytes to serve as outgroups. Both amino acid- and nucleotide-based phylogenies confirmed that embryophyte CYP73 sequences formed a well-defined monophyletic group (Fig. EV1). Charophyte sequences were not associated with the CYP73 group, but instead grouped with other clan 71 CYP families, such as CYP97, CYP98, or CYP701 (Appendix Tab. S1, Fig. EV1). Phylogenetic analyses failed to identify a precise origin of the *CYP73* gene family, or a clearly defined sister family, but suggested that the family arose within the clan 71 (Fig. EV1). To explore the possibility that charophyte proteins might have already a cryptic C4H function, we tested the catalytic activity of the *Klebsormidium nitens* kfl00038_0230 protein that was uncovered as the best hit among non-embryophyte plants (Appendix Tab. S1). In vitro enzyme assay with *K. nitens* recombinant proteins did not result in detectable C4H activity under the tested conditions (Appendix Fig. S1), a result that further supported an emergence of bona fide C4H function in a common ancestor of embryophytes.

To better resolve the phylogenetic relationships within the *CYP73* family, we performed a focused analysis on the 275 *CYP73* sequences only. Apart from the previously reported duplication that occurred in a seed plant common ancestor (Renault et al, 2017b), the topology of the *CYP73* tree reflected embryophyte systematics (Fig. 1C). *CYP73* family was found fairly diversified in bryophytes with for instance four paralogs in the moss

*Physcomitrium patens* (*PpCYP73A48-51*) and three in the liverwort *Marchantia polymorpha* (*MpCYP73A1-3*). In mosses, this diversity could be at least in part explained by an early duplication that occurred after the *Takakia* and *Sphagnum* genera branched off 350 million years ago (Hu et al, 2023) (Fig. 1C). Conservation of C4H function in bryophyte CYP73 proteins was explored through in vitro assays with recombinant proteins representative of the three main groups: PpCYP73A48 (*P. patens*, moss), MpCYP73A1 (*M. polymorpha*, liverwort) and PcCYP73A (*Phaeoceros caroliniaus*, hornwort). PpCYP73A48 and MpCYP73A1 proteins were chosen because they were concurrently investigated in planta in the present study; PcCYP73A protein was selected because it had best full-length RNA-seq scaffold quality in the 1kP database (One Thousand Plant Transcriptomes Initiative, 2019). All three bryophyte CYP73 proteins catalyzed the production of *p*-coumaric acid in vitro, whereas microsomes derived from yeast transformed with the empty vector did not (Fig. 1D). In combination with biochemical data previously obtained in tracheophytes (Mizutani et al, 1997; Renault et al, 2017b), these results indicate that CYP73-catalyzed C4H activity is conserved in embryophytes.

### C4H activity is supported by a conserved arginine residue

To bring further insights into the origin of C4H function in the CYP73 family, we looked for critical residues that support this catalytic activity and that could have underpinned C4H emergence. We used homology modeling to construct the three-dimensional structure of the *P. patens* CYP73A48 enzyme, which had the ability to catalyze C4H activity in vitro (Fig. 1D), based on the crystal structure of *Sorghum bicolor* C4H (Zhang et al, 2020). After docking the heme prosthetic group into the PpCYP73A48 structure, we next docked *t*-cinnamic acid into its active site. The lowest Gibbs free energy pose ($-6.7$ kcal/mol) predicted that position 4 of the *t*-cinnamic acid phenolic ring faced the heme at a distance of 4.3 Å (Fig. 2A), which was consistent with C4H activity. The docking experiment also identified three residues in the F helix of PpCYP73A48—arginine 225 (R225), serine 226 (S226), and glutamine 230 (Q230)—presumably able to form hydrogen bonds with the carboxylic function of *t*-cinnamic acid and to hold it in the proper orientation with respect to the heme reaction center (Fig. 2A). Multiple sequence alignment and corresponding WebLogo visualization showed that these three residues were highly conserved across the 275 CYP73 proteins used for phylogenetic analysis (Fig. 2B). On the contrary, R225, S226, and Q230 residues were not present in charophyte proteins identified in the RBH analysis and in representative members of *A. thaliana* clan 71 (Fig. EV2), confirming the specificity of these residues for the CYP73 family.

To scrutinize the predictions of docking experiments, we focused on the R225 residue of PpCYP73A48, which has a positive charge and can thus strongly interact with the negatively charged carboxyl group of *t*-cinnamic acid. We expanded the evolutionary scope of the experiment by including the previously characterized CYP73A94 and CYP73A92 proteins from the tracheophyte *Brachypodium distachyon* (Renault et al, 2017b) (Fig. 1C). We replaced the R225 residue, or its equivalent in BdCYP73A94 (R241) and BdCYP73A92 (R213), with a non-polar and uncharged alanine residue. Carbon monoxide-induced difference spectra showed that the R > A substitution did not compromise the structural stability of CYP73 proteins (Appendix Fig. S2). In

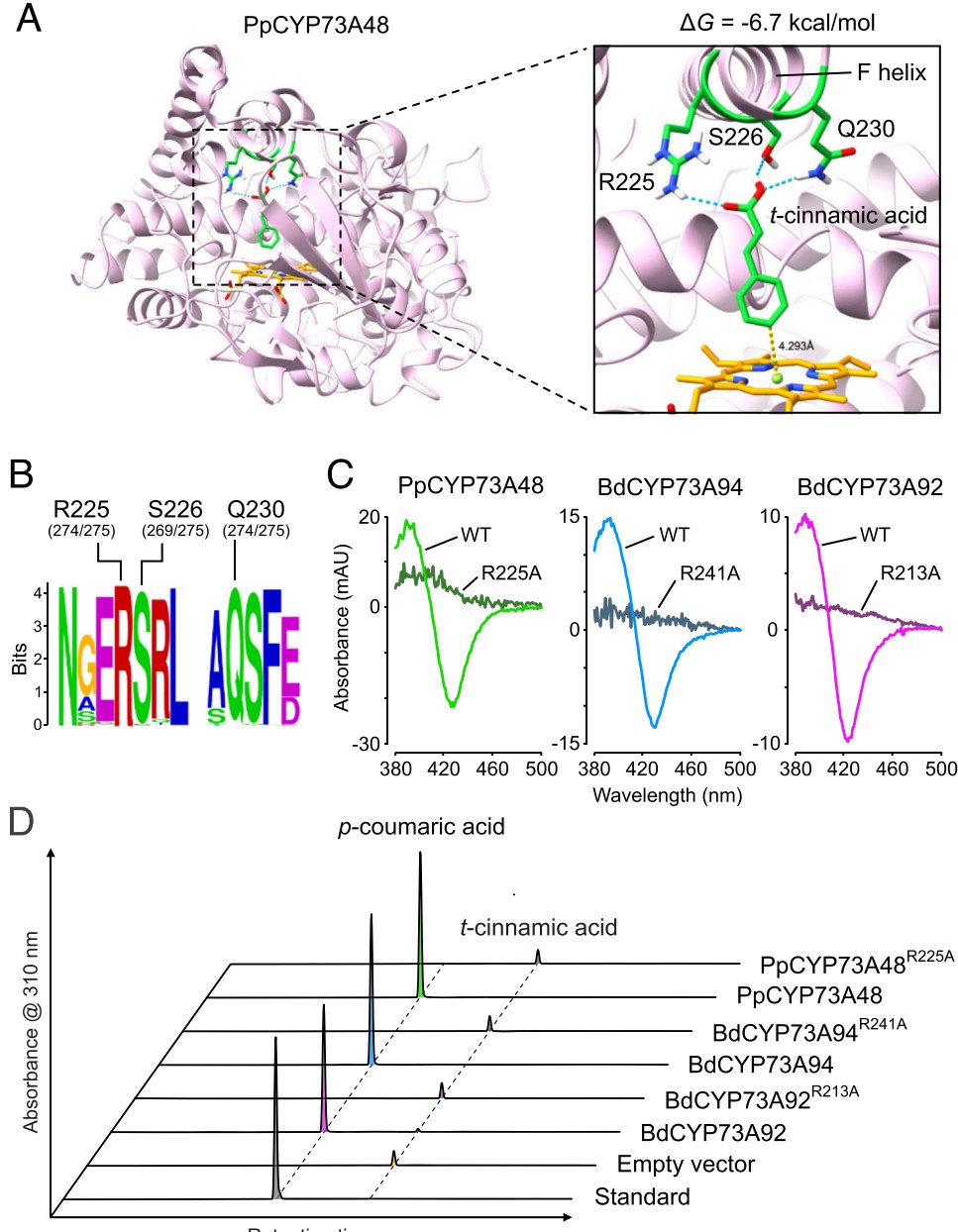

**Figure 2. Structural determinants of *t*-cinnamic acid 4-hydroxylase activity in CYP73.**

(A) Docking of *t*-cinnamic acid in the active site of the *P. patens* CYP73A48 enzyme. Pose corresponding to the lowest Gibbs free energy (ΔG) predicts that R225, S226, and Q230 residues can establish hydrogen bonds with the carboxyl group of *t*-cinnamic acid. In this configuration, *para* position (or position 4) of the phenolic ring faces the heme prosthetic group at a distance of 4.3 Å. (B) WebLogo showing the conservation of R225, S226, and Q230 residues across the 275 CYP73 sequences used for the phylogenetic analyses. For each residue, absolute count is provided between brackets. (C) Representative type I difference spectra showing the loss of *t*-cinnamic acid binding ability of *P. patens* CYP73A48, *B. distachyon* CYP73A94, and *B. distachyon* CYP73A92 enzymes mutated in the R225 residue, or its equivalent. Positively charged arginine residue was replaced with neutral alanine. (D) Representative HPLC-UV chromatograms of in vitro enzyme assays indicating loss of C4H activity in CYP73 enzymes mutated in the R225 residue, or its equivalent. Source data are available online for this figure.

contrast, substrate-induced type I difference spectrum revealed that all three R > A mutated CYP73 proteins lost their ability to bind *t*-cinnamic acid compared to their wild-type counterparts (Fig. 2C). As a result, the R > A CYP73 mutant proteins did not produce *p*-coumaric acid from *t*-cinnamic acid in in vitro assays (Fig. 2D), demonstrating that the conserved R225 residue was a critical determinant of CYP73-catalyzed C4H activity.

## CYP73 deficiency strongly alters moss development

We performed in planta investigation of the *CYP73* genes identified in the moss *P. patens* (Fig. 1C). Scrutiny of public RNA-seq data revealed that only three paralogs were expressed in at least one condition across *P. patens* tissues, with *PpCYP73A48* and *PpCYP73A49* being the most prominent (Fig. 3A). We therefore

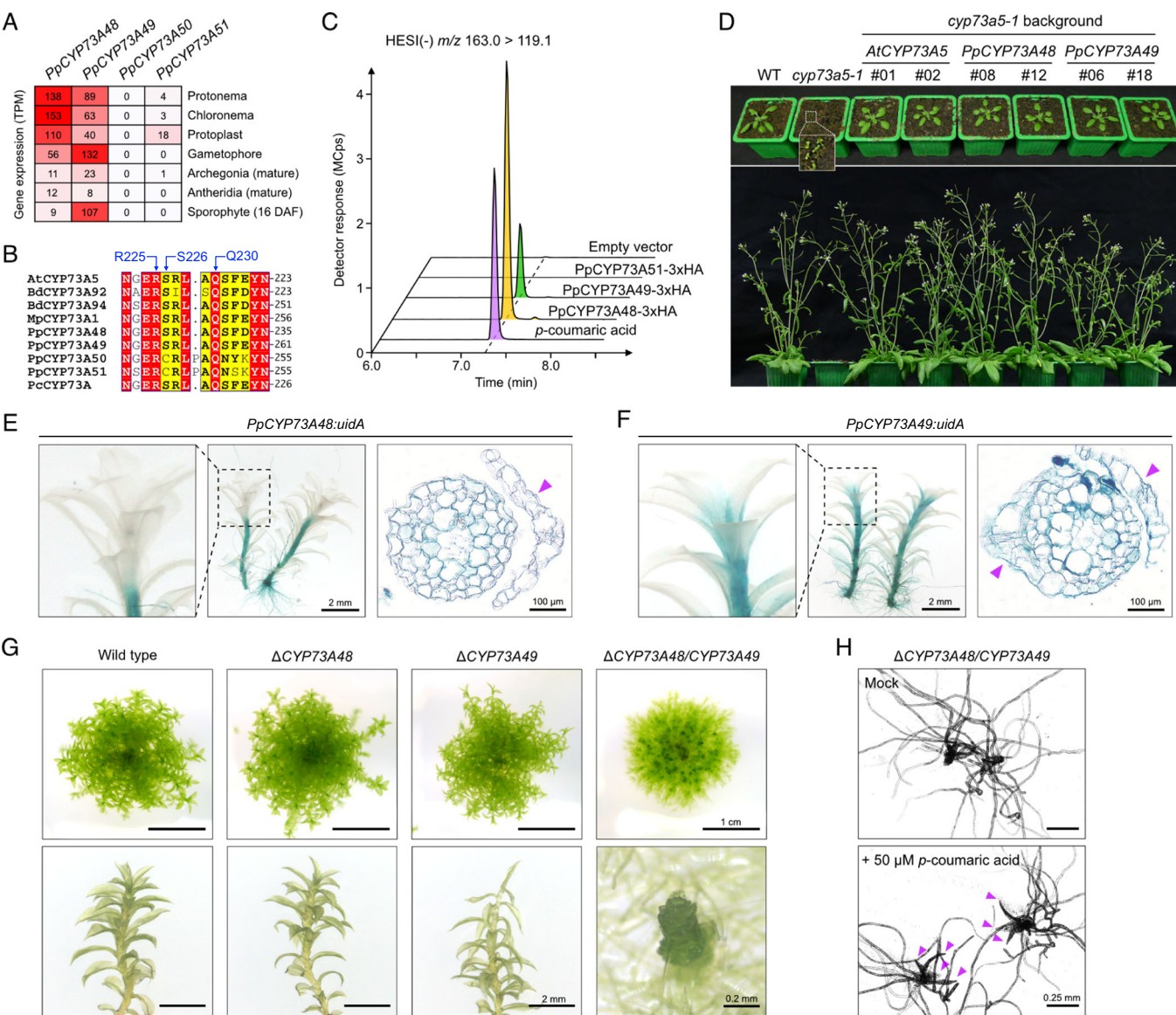

**Figure 3. Functional analysis of *CYP73* genes in the moss *Physcomitrium patens*.**

(A) Expression profiles of the four *P. patens* CYP73 paralogs in various tissues. Data are derived from the CoNekT database (https://evorepro.sbs.ntu.edu.sg; Proost and Mutwil, 2018). DAF, days after fertilization; TPM, transcripts per kilobase million. (B) Angiosperm and bryophyte CYP73 multiple sequence alignment centered on the protein region encompassing *t*-cinnamic acid binding residues identified by docking. At, *Arabidopsis thaliana*, Bd, *Brachypodium distachyon*; Mp, *Marchantia polymorpha*; Pc, *Phaeoceros carolinianus*; Pp, *Physcomitrium patens*. (C) Representative UHPLC-MS/MS chromatograms of in vitro C4H assays performed with recombinant PpCYP73-3xHA tagged proteins. Assays performed with microsomes derived from yeasts transformed with an empty vector were used as negative control. (D) Phenotype of *A. thaliana* 3-week-old (upper panel) and 6-week-old (lower panel) wild type, *cyp73a5-1* mutant and *cyp73a5-1* complemented with *AtCYP73A5*, *PpCYP73A48*, and *PpCYP73A49* coding sequences. Two independent complemented lines are depicted for each gene. (E, F) Representative GUS staining pattern in 2-month-old gametophores of *PpCYP73A48:uidA* (E) and *PpCYP73A49:uidA* (F) reporter lines. For each gene, the central, left, and right pictures show whole gametophores, the apex of a gametophore and a gametophore cross-section, respectively. Magenta arrowheads point to phyllids. (G) Pictures of 2-month-old colonies of wild type, Δ*PpCYP73A48* and Δ*PpCYP73A49* single mutants, and Δ*PpCYP73A48/CYP73A49* double mutant. Close-ups on individual gametophores from each genotype are visible in the lower part. (H) Pictures of 5-week-old Δ*PpCYP73A48/CYP73A49* gametophores grown in low-melting point agarose Knop medium supplemented, or not (control), with 50 μM *p*-coumaric acid. Magenta arrowheads point to developed phyllids. Source data are available online for this figure.

decided to disregard the *PpCYP73A50* paralog, as it displayed no expression and also exhibited substantial variation among the protein region containing the three residues interacting with *t*-cinnamic acid (Fig. 3B). We then performed in vitro assays with PpCYP73A48, PpCYP73A49, and PpCYP73A51 recombinant proteins tagged with 3xHA. While all three proteins were

confirmed to be present in yeast microsomes through Western blot analysis (Appendix Fig. S3), only PpCYP73A48-3xHA and PpCYP73A49-3xHA demonstrated the ability to convert *t*-cinnamic acid into *p*-coumaric acid (Fig. 3C). This observation was further supported in planta by the fact that *PpCYP73A48* and *PpCYP73A49* fully complemented the stunted phenotype of the

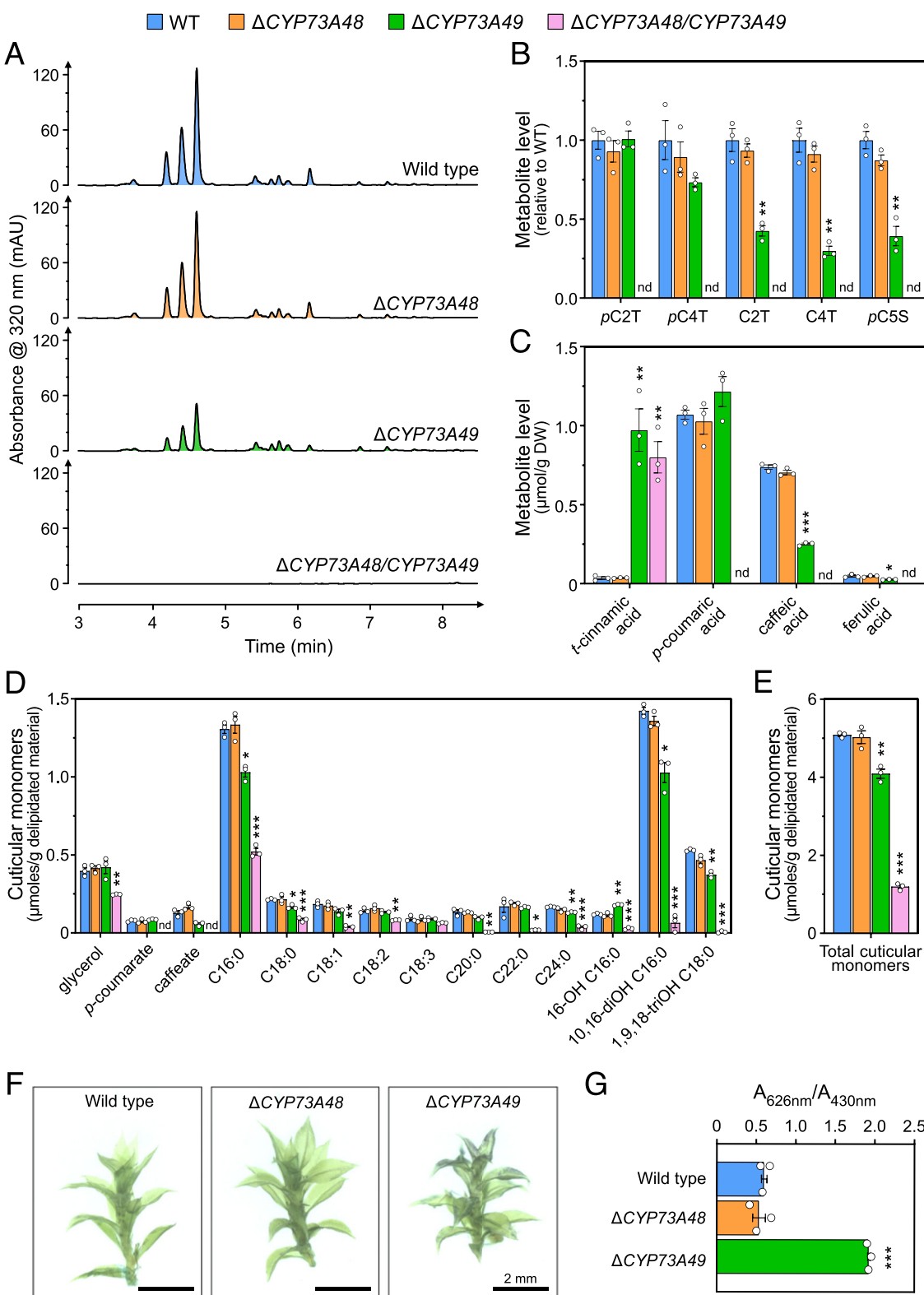

Arabidopsis *cyp73a5-1* mutant (Fig. 3D), whereas no mutant plant rescued with *PpCYP73A51* was ever found.

To refine the analysis of *PpCYP73A48* and *PpCYP73A49* expression patterns, we generated knock-in lines where the STOP codon was replaced with the *uidA* reporter gene. These lines revealed that *PpCYP73A48* and *PpCYP73A49* had overlapping expression domains in gametophores, particularly in the stem (Fig. 3E,F). It was noted, however, that *PpCYP73A49* expression

◀ **Figure 4. Metabolic and physiological characterization of *P. patens* ΔCYP73 mutants.**

(A) Representative UHPLC-UV chromatograms of 2-month-old wild-type, Δ*PpCYP73A48*, Δ*PpCYP73A49*, and Δ*PpCYP73A48/CYP73A49* crude extracts showing the loss of UV-absorbing molecules in the double mutants. (B) Relative levels of threonate and shikimate phenolic esters in 2-month-old gametophores. pC2T, p-coumaroyl-2-O-threonate; pC4T, p-coumaroyl-4-O-threonate; C2T, caffeoyl-2-O-threonate; C4T, caffeoyl-4-O-threonate; pC5S, p-coumaroyl-5-O-shikimate. (C) Quantification of hydroxycinnamic acids from saponified extracts of 2-month-old gametophores. (D) Compositional analysis of the cuticular polymer of 2-month-old gametophores. (E) Total amount of cuticular monomers from gametophores. (F) Representative pictures of 2-month-old wild-type, Δ*PpCYP73A48* and Δ*PpCYP73A49* gametophores after toluidine blue permeability assay. (G) Quantification of permeability to toluidine blue in 2-month-old gametophores. Results in panels (B), (C), (D), (E), and (F) are the mean ± SEM of three independent WT biological replicates and three independent mutant lines. WT versus mutant unpaired *t* test adjusted *P*-value: *$P < 0.05$; **$P < 0.01$; ***$P < 0.001$. nd, not detected. Source data are available online for this figure.

extended further to the apex and in the basal parts of phyllids. We then disrupted both genes individually or concurrently by inserting a selection cassette through homologous recombination, resulting in a loss of corresponding transcripts (Appendix Fig. S4). No strong difference was visible between single mutants and the wild type, except in the Δ*PpCYP73A49* lines where the phyllids at the apex appeared thinner and more fragile (Fig. 3G; Appendix Fig. S5). This latter observation was in accordance with *PpCYP73A49* expression pattern (Fig. 3F). In contrast, the Δ*PpCYP73A48/CYP73A49* double mutant consistently exhibited a very strong developmental phenotype characterized by restricted gametophore growth, phyllid fusion and an overproduction of protonema (Fig. 3G; Appendix Fig. S5). No major changes in the expression of *PpCYP73A48* and *PpCYP73A49* were observed in single mutant backgrounds (Appendix Fig. S6). We then attempted to chemically rescue the double mutant by providing exogenous *p*-coumaric acid. This treatment successfully restored phyllid expansion compared to mock treatment (Fig. 3H; Appendix Fig. S7), indicating that the Δ*PpCYP73A48/CYP73A49* phenotype was, at least in part, caused by a shortage in *p*-coumaric acid, the product of C4H activity.

## CYP73 deficiency leads to phenylpropanoid shortage and cuticle defect in moss

Next, we examined the metabolic consequences of CYP73 deficiency in the moss. UHPLC-UV fingerprints of Δ*PpCYP73* mutant crude extracts revealed the complete loss of UV-absorbing molecules in double mutants (Fig. 4A), the major ones being hydroxycinnamate esters as reported before (Renault et al, 2017a; Kriegshauser et al, 2021). We also noticed a moderate decrease in the main peaks in Δ*PpCYP73A49* mutants as compared to Δ*PpCYP73A48* and the wild-type plants (Fig. 4A). These findings were confirmed through targeted UHPLC-MS/MS analysis of diagnostic phenylpropanoids in both crude and saponified extracts (Fig. 4B,C). None of the targeted phenylpropanoids downstream of the C4H enzymatic step was detected in metabolic extracts from Δ*PpCYP73A48/CYP73A49* double mutants, whereas we observed concomitant accumulation of *t*-cinnamic acid, the C4H substrate, in saponified extracts (Fig. 4B,C). These results indicated that simultaneous disruption of *PpCYP73A48* and *PpCYP73A49* was sufficient to comprehensively abolish C4H function. Although no notable changes in metabolic profile of Δ*PpCYP73A48* mutants were observed, a significant decrease in *p*-coumaroyl-5-O-shikimate, caffeoyl-2-O-threonate and caffeoyl-4-O-threonate level was evident in Δ*PpCYP73A49* crude extracts. Corresponding saponified samples showed the accumulation of *t*-cinnamic acid and a corollary reduction in the amount of caffeic acid, while *p*-coumaric

acid level remained unchanged (Fig. 4C). Since *cis*-cinnamic acid, a *t*-cinnamic acid stereoisomer, was suggested to play a role in the developmental abnormalities linked to C4H deficiency in Arabidopsis (Houari et al, 2021), we conducted a targeted analysis of these two compounds in crude extracts. Neither free *t*-cinnamic acid nor *c*-cinnamic acid accumulated in *P. patens* mutants (Appendix Fig. S8).

We then examined how impairing C4H function affected the composition of the *P. patens* gametophore cuticular polymer, as its formation relies on phenylpropanoid precursors (Renault et al, 2017a; Kriegshauser et al, 2021). The analysis of polymer monomers revealed a significant decrease in all aliphatic components, except C18:3, in the Δ*PpCYP73A48/CYP73A49* double mutants compared to wild type (Fig. 4D). This decrease was accompanied by a complete loss of the two phenolic monomers, *p*-coumarate and caffeate. The overall quantity of the cuticular polymer, represented by the sum of individual monomers, dropped by 75% in the double mutants compared to the wild-type level (Fig. 4E). There were no changes in cuticular polymer composition of Δ*PpCYP73A48* single mutants, consistent with the soluble phenylpropanoid analysis (Fig. 4A–E). In contrast, in Δ*PpCYP73A49* single mutants, there were moderate yet statistically significant variations in cuticular polymer composition (Fig. 4D), resulting in a *ca*. 20% decrease in total monomer level compared to wild type (Fig. 4E). As for phenolic monomers, *p*-coumarate remained unchanged in the cuticular polymer of Δ*PpCYP73A49* mutants, while caffeate level was reduced by 50%, although this finding was not statistically supported (adjusted *P*-value = 0.084). To understand the consequences of polymer composition changes on cuticle properties, we performed toluidine blue assays on gametophores of single Δ*PpCYP73* mutants; the stunted growth of double Δ*PpCYP73A48/CYP73A49* gametophores prevented their inclusion in the experiment. As shown in Fig. 4F,G, we observed a significant increase in Δ*PpCYP73A49* gametophore permeability to toluidine blue, indicating a strong defect in cuticle diffusion barrier function despite the moderate cuticle compositional changes (Fig. 4D). This function appeared unaltered in the Δ*PpCYP73A48* mutant as compared to wild type (Fig. 4F,G).

## CYP73 function is conserved in the liverwort *Marchantia polymorpha*

To bring additional evolutionary perspective to the functional analysis of the *CYP73* gene family, we expanded our investigations to a second bryophyte species, the liverwort *M. polymorpha*. We identified the *MpCYP73A1* gene as the primary paralog based on its expression level (Fig. 5A,B). In addition, we previously confirmed

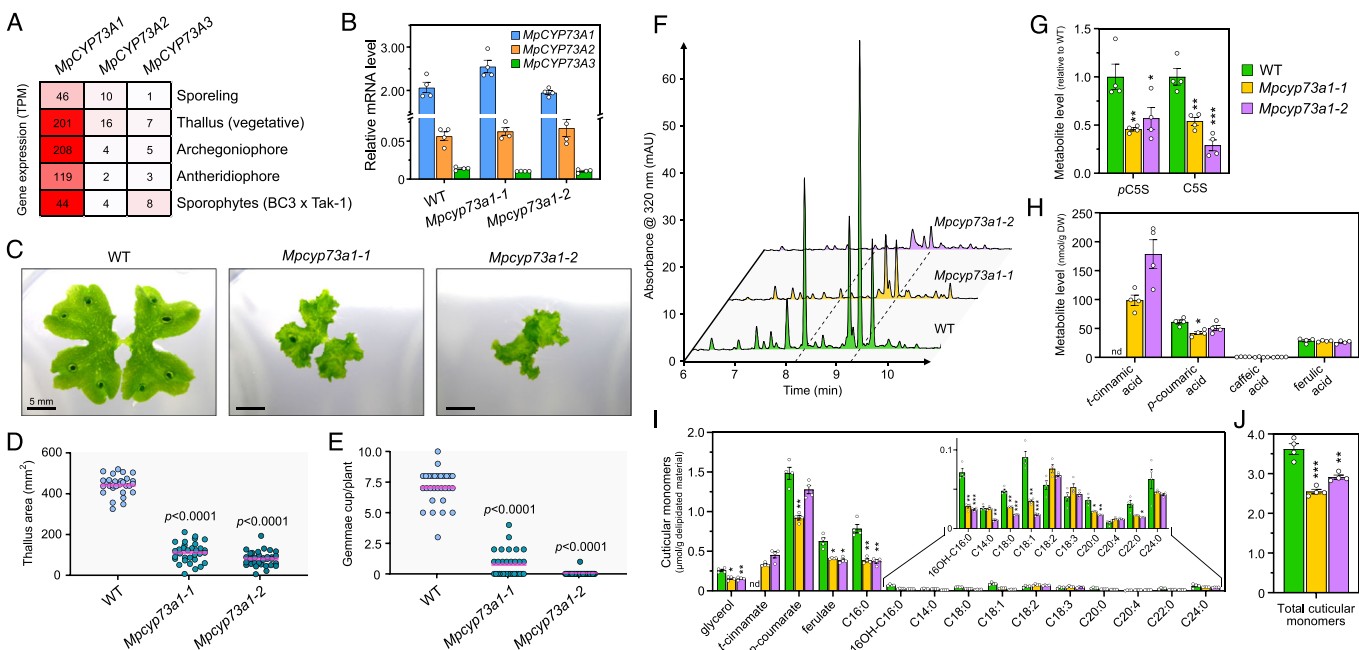

**Figure 5. Developmental and metabolic consequences of CYP73A1 deficiency in *M. polymorpha*.**

(A) Expression profiles of the three *M. polymorpha* CYP73 paralogs in various tissues. Data are derived from the CoNekT database (https://evorepro.sbs.ntu.edu.sg; Proost and Mutwil, 2018), except sporophytes for which data derive from the MarpolBase database (https://marchantia.info/mbex/). (B) RT-qPCR expression analysis of the three *MpCYP73* genes in 3-week-old wild-type, *Mpcyp73a1-1* and *Mpcyp73a1-2* plants. Results are the mean ± SEM of four biological replicates. (C) Representative pictures of 3-week-old wild-type, *Mpcyp73a1-1* and *Mpcyp73a1-2* plants. (D, E) Quantification of thallus area (D) and gemmae cup number (E) in 3-week-old wild-type, *Mpcyp73a1-1* and *Mpcyp73a1-2* plants. Results are the mean (pink trait) of measurements made on 29–32 plants. WT versus mutant one-way ANOVA adjusted *P*-value is displayed. (F) Representative UHPLC-UV chromatograms of 1-month-old wild-type, *Mpcyp73a1-1* and *Mpcyp73a1-2* crude extracts. (G) Relative quantification of shikimate phenolic esters in 1-month-old wild-type, *Mpcyp73a1-1* and *Mpcyp73a1-2* plants. C5S, caffeoyl-5-*O*-shikimate; *p*C5S, *p*-coumaroyl-5-*O*-shikimate. (H) Quantification of hydroxycinnamic acids from saponified extracts of 1-month-old wild-type, *Mpcyp73a1-1* and *Mpcyp73a1-2* plants. (I) Compositional analysis of the cuticular polymer in 1-month-old wild-type, *Mpcyp73a1-1* and *Mpcyp73a1-2* plants. (J) Total amount of cuticular monomers in 1-month-old wild-type, *Mpcyp73a1-1* and *Mpcyp73a1-2* plants. Results in panels (G), (H), (I), and (J) are the mean ± SEM of four independent biological replicates for each genotype. WT versus mutant unpaired *t* test adjusted *P*-value: *$P < 0.05$; **$P < 0.01$; ***$P < 0.001$. nd, not detected. Source data are available online for this figure.

that the corresponding protein catalyzes C4H activity in vitro (Fig. 1D). We therefore produced two independent CRISPR mutants of the *MpCYP73A1* gene, using two distinct protospacers located in the first exon (Appendix Fig. S9). The resulting *Mpcyp73a1-1* and *Mpcyp73a1-2* alleles featured 12 and 37 nucleotide deletions, respectively, leading to four amino acid loss (residues 94–97) and a premature STOP codon (position 150) at the protein level. The impact of these mutations on *M. polymorpha* development was significant, leading to a dramatic reduction in both the thallus area and number of gemmae cups per plant compared to the wild type (Fig. 5C–E). Noteworthy, disrupting *MpCYP73A1* function did not affect the expression of the two other *MpCYP73* paralogs, which remained 28- to 243-times lower in the thallus (Fig. 5B).

We went on with the metabolic analysis of *Mpcyp73a1* mutants. Through UHPLC-UV fingerprints of crude metabolic extracts, we observed a decrease in main peaks in mutants as compared to wild type (Fig. 5F). Targeted UHPLC-MS/MS analysis demonstrated *ca.* 50% reduction in the two essential intermediates of the phenyl-propanoid pathway, *p*-coumaroyl-5-*O*-shikimate and caffeoyl-5-*O*-shikimate (Kriegshauser et al, 2021), in *Mpcyp73a1* CRISPR lines (Fig. 5G). As with *P. patens*, we found no evidence of free *c*-cinnamic acid accumulation in *Mpcyp73a1* mutant lines, although

*t*-cinnamic acid was present (Appendix Fig. S8). Saponification of the crude extracts followed by targeted analysis revealed no significant changes in hydroxycinnamic acids, except *t*-cinnamic acid that appeared in the *Mpcyp73a1* mutants (Fig. 5H). These results suggest that the major UV-absorbing peaks in *M. polymorpha* crude extracts, and which decrease in mutant lines (Fig. 5F), were not hydroxycinnamic acid derivatives but rather belonged to other phenylpropanoid classes (e.g., auronidins) (Berland et al, 2019). Phenylpropanoid biosynthesis was not completely abolished in *Mpcyp73a1* mutants, probably due to genetic redundancy since two additional, lowly expressed *MpCYP73A* paralogs occur in this species (Fig. 5A,B).

We extended the metabolic characterization of *Mpcyp73a1* mutants to the compositional analysis of their cuticular polymer. This analysis uncovered phenolics as the most abundant monomers in *M. polymorpha* (Fig. 5I). Both mutants consistently accumulated *t*-cinnamate in their polymer and featured a *ca.* 35% reduction in ferulate compared to the wild type. Impairing *MpCYP73A1* function also caused alterations in the aliphatic monomer composition of the cuticular polymer, including decrease in glycerol, palmitate (C16:0), and ω-hydroxypalmitate (16OH-C16:0) (Fig. 5I). Overall, the cuticular polymer quantity was significantly reduced, by up to 25%, in *Mpcyp73a1* mutant lines (Fig. 5J).

## A selective C4H inhibitor consistently impairs development and phenylpropanoids in embryophytes

Finally, we performed pharmacological experiments using the selective inhibitor piperonylic acid (PA) to disrupt C4H function in planta (Schalk et al, 1998; Naseer et al, 2012). PA treatment accurately replicated in *P. patens* wild-type plants the developmental phenotype observed in Δ*PpCYP73A48/CYP73A49* double mutants, displaying stunted gametophores and protonema outgrowth (Figs. 3G and 6A). Similarly, we observed a dramatic inhibition by PA of *M. polymorpha* thallus expansion, which resembled an aggravation of *Mpcyp73a1* single mutant phenotypes (Figs. 5C and 6A). Based on these findings, we reasoned that PA treatment could serve as a substitute for *CYP73* gene inactivation, allowing to investigate C4H function in a wider range of streptophyte species. We therefore subjected the fern *Ceratopteris richardii* (tracheophyte), the hornwort *Anthoceros agrestis* (bryophyte) and the charophyte *Klebsormidium nitens* to PA treatment. *K. nitens* has no *CYP73* gene and thus virtually lacks PA target enzyme (Fig. 1B; Appendix Tab. S1, Fig. EV1). PA treatment led to developmental changes in *C. richardii*, where sporophyte growth was retarded, and *A. agrestis*, where thallus expansion was inhibited and associated with rounded and thickened edges (Fig. 6A). In contrast, no visible deviation in *K. nitens* development was observed after two weeks in PA-supplemented medium (Fig. 6A). We then investigated the metabolic consequences of PA treatment in the five streptophyte species. UHPLC-UV fingerprints of crude metabolic extracts revealed a consistent decrease in the major UV-absorbing peaks in embryophyte chromatograms (i.e., *C. richardii*, *P. patens*, *M. polymorpha*, and *A. agrestis*) (Fig. 6B), indicating that phenylpropanoid biosynthesis was impaired. *K. nitens* was notably devoid of molecules absorbing at 320 nm under the tested conditions, supporting the absence of a phenylpropanoid pathway in this charophyte algae (Fig. 6B). We confirmed UV fingerprint data by the targeted analysis of the two diagnostic phenylpropanoid compounds *p*-coumaroyl-5-*O*-shikimate and caffeoyl-5-*O*-shikimate. This analysis revealed a significant decrease in both molecules in embryophytes after PA treatment, except in *M. polymorpha* where *p*-coumaroyl-5-*O*-shikimate level was unchanged (Fig. 6C). The latter observation may result from an aspecific signal since *p*-coumaric acid, the corresponding free hydroxycinnamic acid, was significantly decreased in saponified extracts of PA-treated *M. polymorpha* plants (Appendix Tab. S2). Neither *p*-coumaroyl-5-*O*-shikimate nor caffeoyl-5-*O*-shikimate was detected in the metabolic extracts of *K. nitens* (Fig. 6C).

## Discussion

Based on both computational and experimental data, our updated evolutionary analysis provides evidence that C4H activity emerged in an embryophyte progenitor together with the *CYP73* gene family. These findings resonate with previous studies showing that potential *CYP73* orthologs are exclusive to embryophytes (Renault et al, 2017b; de Vries et al, 2017, 2021). Whether this evolutionary scenario will stand valid in light of upcoming, new genomic data from streptophyte algae remains an open question. Nonetheless, the evolutionary pattern of the *CYP73* family diverges from that of other genes within the phenylpropanoid pathway, ranging from *PAL* to *CYP98*, for which

distant homology is evident in charophytes (de Vries et al, 2017, 2021; Renault et al, 2019). The seemingly sudden and delayed emergence of the *CYP73* gene family thus raises questions about the evolutionary path and mechanisms that gave birth to this gene family. It also suggests that *CYP73* emergence was instrumental in founding the canonical phenylpropanoid pathway in an embryophyte ancestor, in line with the often observed contribution of new CYP families to creating novel metabolic pathways (Mizutani and Ohta, 2010; Hansen et al, 2021). Moreover, previous molecular evolution analysis underscored the consistent trend of strong purifying selection acting on *CYP73* genes across both tracheophytes and bryophytes, hinting at the fact they fulfilled essential function from the early stages of land plant evolution (Renault et al, 2017b).

Together with data from the study of the Arabidopsis *CYP73A5* gene (Schilmiller et al, 2009), our functional analysis of bryophyte *CYP73* genes establishes the C4H step as a critical and evolutionarily conserved element of the canonical phenylpropanoid pathway. The recruitment and fixation of C4H within embryophytes delineates an intriguing facet, particularly when considering the potential of some plant enzymes to establish an alternative route to *p*-coumaric acid. A prominent example are the bifunctional phenylalanine/tyrosine ammonia-lyases (PTAL), which derive from to the ubiquitous PAL enzyme family. In the monocot *Brachypodium distachyon*, PTAL bypasses C4H and contributes to nearly half of lignin production through L-tyrosine (Barros et al, 2016). However, PTAL are mainly restricted to grasses, and *B. distachyon* retained three *CYP73* genes that encode functional C4H enzymes (Renault et al, 2017b). The precise benefits of positioning and conserving C4H as a central module in the phenylpropanoid pathway is elusive. Exploration of catalytic properties revealed the high affinity ($K_m < 10\,\mu M$) and relatively slow turnover rates of C4H enzymes across various species (Renault et al, 2017b; Nedelkina et al, 1999; Pierrel et al, 1994; Wu et al, 2018). Furthermore, it was observed that C4H substrate preference is constrained, being capable of using only *t*-cinnamic acid mimics as substrates (Schalk et al, 1998, 1997; Chen et al, 2007; Pierrel et al, 1994). Thus, the catalytic attributes of C4H potentially prevent metabolic derailment that could result from enzyme promiscuity, while concurrently offering the potential for efficient control over metabolic flux. This perspective is further supported by the documented role of the ER membrane-bound C4H as a nucleation point for a phenylpropanoid metabolon (Bassard et al, 2012; Achnine et al, 2004), facilitating the recruitment of other CYP and soluble enzymes within close proximity for the effective and dynamic channeling of metabolic intermediates.

Our work demonstrates that perturbing C4H function, either via *CYP73* gene inactivation or inhibitor treatment, had a significant impact on bryophyte development. This aligns with the findings obtained in the tracheophyte Arabidopsis, where interfering with the five first steps of the phenylpropanoid pathway consistently led to lignin alteration and growth reduction (Schilmiller et al, 2009; Franke et al, 2002; Hoffmann et al, 2004; Huang et al, 2010; Li et al, 2015). Four hypotheses were proposed to explain this phenomenon, ranging from impaired water transport due to xylem collapse, negative feedback through a cell wall surveillance system, and over- or under-accumulation of phenylpropanoid derivatives (reviewed in Muro-Villanueva et al, 2019). Whether the developmental alterations we observed in bryophytes, which lack lignified vascular tissues, falls under some of these hypotheses is an interesting question. It's worth noting that the impact of impeding C4H function in bryophytes

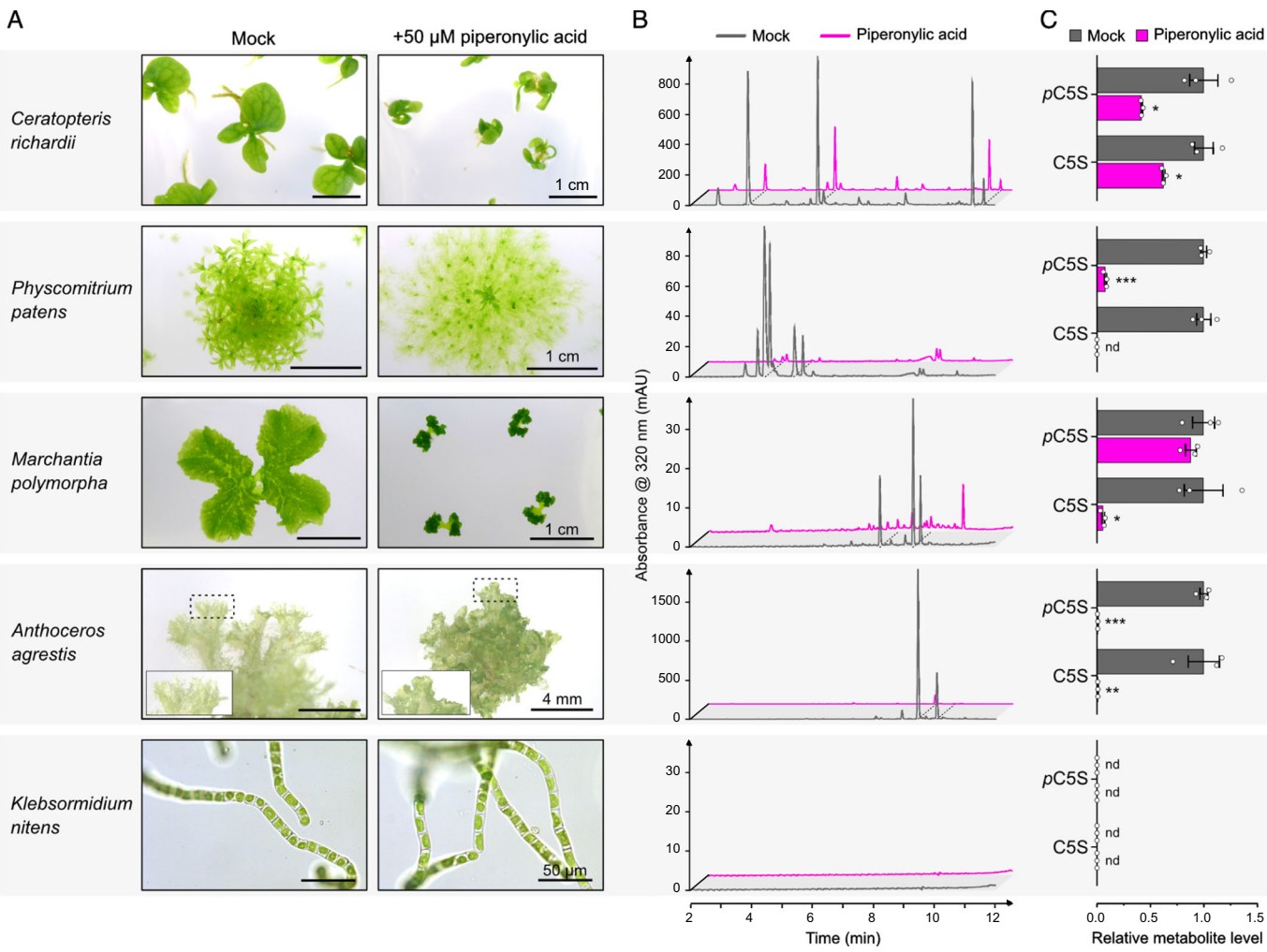

**Figure 6. Developmental and metabolic effects of the C4H inhibitor piperonylic acid in streptophytes.**

Unless otherwise stated, wild-type plants were treated with 50 μM piperonylic acid (PA) or corresponding mock (0.05% DMSO) on agar plates. *C. richardii* (fern) was treated for three weeks after transfer of 2-week-old sporophytes. *P. patens* was treated for one month after transfer of individual gametophores. *M. polymorpha* was treated for three weeks after transfer of individual gemmae. *A. agrestis* was treated for two months after transfer of thallus pieces. *K. nitens* (charophyte, Klebsormidiophyceae) was treated in liquid cultures for 2 weeks after subculturing (1:10 ratio). (**A**) Pictures of plants after PA or mock treatment. (**B**) Representative UHPLC-UV chromatograms of crude metabolic extracts of mock- and PA-treated plants. *P. patens* metabolic analysis was performed on plants grown in liquid culture to facilitate tissue collection. (**C**) Relative quantification of shikimate phenolic esters in mock- and PA-treated plants. C5S, caffeoyl-5-*O*-shikimate; *p*C5S, p-coumaroyl-5-*O*-shikimate. Results are the mean ± SEM of three independent biological replicates for each condition. Mock versus PA unpaired *t* test adjusted *P*-value: *P < 0.05; **P < 0.01; ***P < 0.001. nd, not detected. Source data are available online for this figure.

extends beyond mere growth reduction. It profoundly alters the developmental plan, leading for instance in *P. patens* to phyllid expansion failure and organ fusion symptoms, as observed in mutants of downstream steps (Renault et al, 2017a; Kriegshauser et al, 2021). Interestingly, this strong developmental phenotype could be mitigated by supplying exogenous *p*-coumaric acid. This suggests that the observed phenotypes were unlikely due to hyperaccumulation of *t*-cinnamic acid derivatives. As proposed for other *P. patens* phenylpropanoid mutants (Renault et al, 2017a; Kriegshauser et al, 2021), phenylpropanoid shortage might account for the developmental anomalies in C4H-impaired bryophytes. This hypothesis could be linked to cuticle defects, which might function, in absence of lignin and suberin, as a structural element in bryophytes, contributing to tissue scaffolding. This perspective does not necessarily contradict the two remaining hypotheses—feedback control through cell wall monitoring

system or impaired water transport. However, the latter hypothesis appears less substantiated, as neither growing *P. patens* Δ*PpCYP73* mutants in liquid culture nor embedding them in agarose medium could rescue their stunted phenotype. Further investigations will be necessary to elucidate the factors driving the developmental consequences of C4H deficiency in bryophytes and to determine whether these mechanisms are conserved across embryophytes.

## Methods

### Homolog search and phylogenetic reconstruction

Search for *CYP73* homologs involved a reciprocal best hits (RBH) method with AtCYP73A5 protein sequence as the initial query (see

Appendix Tab. S1 for RBH output). Briefly, we conducted a forward BLASTp search across 20 Viridiplantae genomes. Next, for each species, the top hit based on the bit-score was used to perform a reverse BLASTp search against the *Arabidopsis thaliana* genome. The presence of a potential *AtCYP73A5* homolog was considered positive when AtCYP73A5 was identified as the best hit in the reverse search. We updated our previously published *CYP73* phylogeny (Renault et al, 2017b) with 72 new sequences retrieved from recently released genomes or through an extended search in the 1kP transcriptome dataset (One Thousand Plant Transcriptomes Initiative, 2019), resulting in a total of 275 CYP73 sequences. Additional sequences were retrieved from *P. patens*, *A. thaliana* and charophyte genomes by tBLASTn search using AtCYP73A5 protein as query. Applying a bit-score cut-off >100, this search identified 271 CYP sequences, which served as outgroups for subsequent phylogenies. Multiple sequence alignment were performed on protein sequences using the MUSCLE algorithm (Edgar, 2004). Ambiguous regions in obtained alignments were masked using Gblocks (Castresana, 2000) implemented via Seaview (Gouy et al, 2010). For nucleotide phylogenies, protein alignments were reverse-translated to their initial nucleotide sequence prior to phylogenetic analysis (full alignments available in Dataset EV1 and Dataset EV2). Maximum-likelihood phylogenies were reconstructed from the Gblocks alignments using IQ-TREE2 v2.2.2.6 (Minh et al, 2020). The reconstruction employed the following command line: "./iqtree2 -s Gblock_alignment.fst --alrt 1000 -B 1000", which implements ModelFinder for model selection and computes SH-aLRT tests and ultrafast bootstrap with 1000 pseudo-replicates to determine branch support (tree files available in Datasets EV3, EV4 and EV5). Phylogenetic trees were visualized and edited in FigTree v1.4.4 (http://tree.bio.ed.ac.uk/software/figtree/) or iTOL v6 (https://itol.embl.de), and finally annotated with Affinity Designer.

## Homology modeling and docking experiments

Three-dimensional structure of transmembrane segment-free PpCYP73A48 protein (residues 46–517) was built using the SWISS-MODEL server homology modeling pipeline (Waterhouse et al, 2018). Both BLAST and HHblits were used to search for the best template in SWISS-MODEL library. Models were built based on the target-template alignment using ProMod3 v3.1.1 and *Sorghum bicolor* C4H1 structure (pdb: 6VBY) as template. The resulting structure of PpCYP73A48 exhibited a good structural alignment with SbC4H1, with a global QMEANDisCo score of 0.87 ± 0.05. Heme prosthetic group was first docked into reconstructed PpCYP73A48 3D model using Autodock Vina (Trott and Olson, 2010) in flexible mode, allowing geometry of cysteine 459 to change. Resulting heme-containing PpCYP73A48 3D model (pdb file available in Dataset EV6) was subsequently used for *t*-cinnamic acid docking via Autodock Vina in rigid mode. A 40 Å × 40 Å × 40 Å search box positioned above the heme was used. Ligand and receptor files required for docking experiments were prepared with AutodockTools 1.5.7 (https://ccsb.scripps.edu/mgltools/). Docking results were visualized with ChimeraX software (Pettersen et al, 2021).

## Generation of yeast expression plasmids

Coding sequences (CDS) of *Physcomitrium patens CYP73A48* (*Pp3c4_21680V3.1*), *CYP73A49* (*Pp3c25_10190V3.1*) and *CYP73A51* (*Pp3c3_17840V3.1*) were PCR-amplified from *P. patens*

Gransden cDNA and cloned into the yeast expression plasmid pYeDP60 by restriction/ligation using *Bam*HI/*Kpn*I (CYP73A48 and CYP73A51) or *Sma*I/*Kpn*I (CYP73A49) restriction sites. *Marchantia polymorpha CYP73A1* (*Mp6g00020.1*) coding sequence was PCR-amplified from *M. polymorpha* Tak-1 cDNA using Gateway-compatible primers. Yeast codon-optimized *Phaeoceros carolinianus CYP73A* (*PcCYP73A*; RXRQ_scaffold_2140006) and native *Klebsormidium nitens kfl00038_0230* coding sequences were synthesized as double-strand DNA fragments (gBlock, Integrated DNA Technologies) containing Gateway *att*B1 and *att*B2 extensions (primer sequences available in Appendix Tab. S3). After generation of pENTRY plasmids by BP recombination, *MpCYP73A1*, *PcCYP73A*, and *kfl00038_0230* CDS were shuttled to a Gateway version of pYeDP60 by LR recombination. For 3xHA tagging of CYP73 proteins, STOP codon-free CDS were PCR amplified using Gateway-compatible primers and cloned into pDONR207 by BP recombination. CDS were transferred to a modified pAG425GAL-ccdB-3xHA yeast expression plasmid (Alberti et al, 2007), in which the *LEU2* auxotrophic marker was replaced by *ADE2*. Plasmids containing yeast-optimized CDS of *Brachypodium distachyon CYP73A92* and *CYP73A94* were described previously (Renault et al, 2017b). For expression of CYP73 mutated proteins, site-directed mutagenesis was performed on pYeDP60 plasmids by overlapping PCR, using primers containing the desired mutations. To reduce background originating from methylated template plasmids, PCR reactions were treated with *Dpn*I prior to *E. coli* transformation.

## Production of recombinant CYP73 proteins in yeast

Yeast expression plasmids harboring *CYP73* CDS were introduced into the *Saccharomyces cerevisiae* WAT11 strain (Urban et al, 1997). Procedures for recombinant CYP expression and microsome purification were the same as in Liu et al (2016). Total microsomal protein concentration was determined according to the Qubit™ protein assay kit (ThermoFisher Scientific). Recombinant CYP proteins were quantified by type II differential spectrophotometry using reduced cytochrome P450 and carbon monoxide (Guengerich et al, 2009). Expression of CYP73-3xHA tagged proteins was analyzed by Western blot. To this end, microsomal proteins were denatured for 5 min at 65 °C in Laemmli buffer containing DTT as reducing agent. Proteins were quantified by the amido black protein assay with bovine serum albumin as reference. Ten micrograms of denatured microsomal proteins were separated by SDS-PAGE using 10% polyacrylamide gel (Mini-PROTEAN TGX precast gel, Biorad) and tris/glycine buffer. After electrophoresis, proteins were transferred on Immobilon-P PVDF membrane. HA-tagged proteins were detected using an anti-HA antibody (1/10,000 dilution; Sigma #H9658) followed by incubation with a peroxidase-conjugated goat anti-mouse IgG antibody (1/5000 dilution; Invitrogen # G21234) and finally revealed by chemiluminescence (Clarity Western ECL, Biorad) on a Fusion-FX imager (Vilber).

## In vitro enzyme assays

Standard assay for cinnamic acid 4-hydroxylase activity was performed in a 100 µL reaction containing 50 mM potassium phosphate buffer (pH 7.4), 150 µg microsomal proteins (~5 µl microsomal preparations), 100 µM *trans*-cinnamic acid (from a

10 mM stock solution prepared in DMSO) and 500 μM NADPH. The reaction was initiated by addition of NADPH and incubated at 28 °C in the dark for 30 min. Reaction was stopped by the addition of 100 μl methanol followed by thorough agitation. Samples were centrifuged for 10 min at $15,000 \times g$, 4 °C to pellet microsomes. Supernatant was recovered and used for UHPLC-MS/MS analysis. For in vitro assay with mutated CYP73 enzymes, same procedures were followed except that 5 pmoles P450 enzyme per assay were used, and samples were analyzed by HPLC-UV. The *t*-cinnamic acid binding properties of CYP73 recombinant proteins were assayed spectrophotometrically by monitoring the low-spin to high-spin transition upon substrate binding in the active site (i.e., type I spectra). To this end, absorbance difference in the 380–500 nm range was measured with 150 nM recombinant CYP73 proteins and 100 μM *t*-cinnamic acid.

## Plant material and growth conditions

We used in this study *Physcomitrium patens* Gransden (Rensing et al, 2008), *Marchantia polymorpha* Tak-1 (Bowman et al, 2017), *Anthoceros agrestis* Bonn (Li et al, 2020), *Ceratopteris richardii* Hnn (Marchant et al, 2022) and *Klebsormidium nitens* NIES-2285 (Hori et al, 2014) strains. The Arabidopsis *cyp73a5-1* mutant was previously characterized (Renault et al, 2017b). *P. patens* and *A. agrestis* were grown axenically in Knop medium as described before (Renault et al, 2017a). *M. polymorpha* was grown axenically in half-strength Gamborg B5 medium (Duchefa, #G0209) containing 0.5 g/L MES as pH buffering agent. Gamborg B5 medium was adjusted to pH 5.5 with KOH and solidified with 12 g/L agar (#4807, Roth). *C. richardii* plants were grown axenically from spores to sporophytes on C-medium agar plates as previously described (Hickok and Warne, 1998). *K. nitens* was grown in half-strength Gamborg B5 liquid medium containing 0.5 g/L HEPES (pH adjusted to 7.0 with KOH). Agar plate cultures were performed in 93 × 21 mm petri plates filled with 30 mL of medium. Liquid cultures were performed in 500 mL Erlenmeyer flasks filled with 200 mL liquid medium and sealed with C-type Silicosen® stoppers. Liquid cultures were constantly agitated at 130 rpm on an orbital shaker. Arabidopsis was grown on soil in 7 cm square pots. Plants were kept under a 22/18 °C, 16 h/8 h light/dark regime. Light was provided by 20W/840 white cool LED tubes (Philips) at an intensity of 50 μmol/m²/s for bryophyte and fern species, and 100 μmol/m²/s for Arabidopsis.

## Generation of *P. patens* transgenic lines

Δ*PpCYP73s* knock-out mutants were generated via homologous recombination-mediated gene disruption as described previously (Kriegshauser et al, 2021). *CYP73A48* and *CYP73A49* disruption constructs were excised from vector backbone by *Bam*HI and *Kpn*I digestion, respectively. The Δ*PpCYP73A48/CYP73A49* double mutants were produced by disrupting the *PpCYP73A49* gene in the Δ*PpCYP73A48* #23 mutant background. For *uidA* reporter lines, two genomic regions for homologous recombination framing the STOP codon were PCR-amplified from genomic DNA and assembled with the *uidA* reporter gene following the same procedures as in Kriegshauser et al (2021). The *CYP73A48:uidA* and *CYP73A49:uidA* constructs were excised from vector backbone by *Nhe*I digestion. 25 μg of excised fragment were used for protoplasts transfection. Since *CYP73:uidA* constructs do not

contain a selection marker, it was co-transfected with the pRT101 plasmid containing the *NPTII* selection cassette (Girke et al, 1998). Transformants were selected on Knop plates supplemented with 25 mg/L geneticin (Δ*PpCYP73A48* mutants and *PpCYP73A:uidA* lines) or 10 mg/L Hygromycin B (Δ*PpCYP73A49* mutants). Transgenic lines were molecularly characterized as described previously (Kriegshauser et al, 2021).

## Arabidopsis *cyp73a5-1* mutant trans-complementation

Coding sequences of *CYP73A48*, *CYP73A49*, *CYP73A51*, and *CYP73A5* were transferred by LR recombination into the Gateway pCC1061 binary vector, which contains a 2977 bp promoter fragment from Arabidopsis *CYP73A5* gene (Weng et al, 2011). Obtained expression plasmids were introduced into *Agrobacterium tumefaciens* C58c1 strain and used to transform heterozygous *cyp73a5-1* plants by the floral dip method (Clough and Bent, 1998). Transformants were selected based on their resistance to kanamycin. T1 plants homozygous for the *cyp73a5-1* allele were identified by PCR (primer sequences available in Appendix Tab. S3) and further confirmed by the full resistance of T2 progeny to sulfadiazine. Experiments were performed with T3 plants homozygous for both the mutant allele and the trans-complementation construct.

## GUS staining

Plant tissues were vacuum infiltrated during 10 min with a GUS solution containing 50 mM potassium phosphate buffer pH 7.0, 0.5 mM ferrocyanide, 0.5 mM ferricyanide, 0.1% Triton X-100 and 0.5 mg/mL X-Gluc, and incubated at 37 °C for 4.5 h. Chlorophyll was removed by washing tissues three times in 70% ethanol. For gametophore stem cross-section, cleared gametophores in 70% ethanol were embedded in Paraplast embedding medium (Electron Microscope Sciences). 25 μm sections were prepared with a Leica RM2155 microtome and were imaged on a Leica DMRB microscope.

## Chemical complementation with *p*-coumaric acid

Protoplasts from *P. patens* wild type and Δ*PpCYP73A48/CYP73A49* mutant were embedded in low-melting point agarose as described before (Wiedemann et al, 2018). After three days, regeneration solution that overlaid the solidified film was changed to Knop medium supplemented 50 μM *p*-coumaric acid. Mock treatment was performed by supplementing Knop medium with 0.1% ethanol. Regeneration of protoplasts was performed in standard growth conditions and monitored over five weeks.

## Soluble metabolite extraction and analysis

Tissue collection, lyophilization and grinding were performed as described before (Kriegshauser et al, 2021). Metabolites were extracted from lyophilized plant material using a methanol:chloroform:water protocol as described previously (Kriegshauser et al, 2021). Briefly, 500 μl methanol were added to 10 mg lyophilized plant material. Samples were agitated for 1 h at 1500 rpm at room temperature prior to addition of 250 μl chloroform. After agitation for 5 min, phase separation was induced by addition of 500 μl water

followed by vigorous agitation and centrifugation ($15,000 \times g$, 4 °C, 15 min). Supernatants were recovered and constituted the crude metabolic extracts. To release free hydroxycinnamic acid (HCAA) from corresponding soluble esters, crude metabolic extracts were saponified. To this end, 200 μl of extract were dried in vacuo, resuspended in 200 μl 1 M NaOH and incubated for 2 h at 30 °C under 1000 rpm agitation. NaOH was neutralized with 33.3 μl 6 M HCl and HCAA were extracted twice with 1 mL ethyl acetate. Pooled organic phases were dried in vacuo and resuspended in 200 μl 50% methanol prior to analysis. Unless otherwise stated, metabolites were separated and detected on a Dionex UltiMate 3000 UHPLC (ThermoFisher Scientific) system coupled to an EvoQ Elite LC-TQ (Bruker) mass spectrometer as reported before (Kriegshauser et al, 2021). Molecules were ionized in positive or negative mode via a heated electrospray ionization source (HESI, Bruker) and detected by specific multiple reaction monitoring (MRM) methods (Appendix Tab. S4). Concurrently to MS/MS data acquisition, UV-absorbance profiles were recorded with an Ultimate 3000 photodiode array detector operated in the 200–400 nm range (ThermoFisher Scientific). UV data were analyzed with Compass Data Analysis software (Bruker). In the case of mutated CYP73 enzyme assays, metabolites were analyzed by HPLC-UV on a Alliance 2695 chromatographic system (Waters) coupled to a photodiode array detector (PDA 2996; Waters) as described before (Kriegshauser et al, 2021).

## Determination of cuticular polymer composition

*P. patens* cuticular polymer composition, including glycerol, was determined from 2-month-old, lyophilized gametophores grown in liquid culture as previously reported (Philippe et al, 2016). For *M. polymorpha*, same procedures were followed except that heptadecanoic acid (C17:0) and ribitol were used as internal standards. *M. polymorpha* samples were analyzed using gas chromatography (GC; Agilent GC8890) coupled with a time-of-flight mass spectrometer (TOFMS; Leco Pegasus BT2). One microliter of derivatized monomers was injected on a HP-5MS Ultra Inert column (30 m × 0.25 mm × 0.25 μm; Agilent) in 100:1 split mode. The temperature gradient was set as follows: 120 °C for 1 min, 120 °C to 340 °C at 10 °C/min, and 340 °C for 3 min. Helium was used as the carrier gas at a flow rate of 1 mL/min. Transfer line and source temperatures were maintained at 250 °C. Analytes were ionized and fragmented by electronic impact at 70 eV. Data were acquired over a *m/z* 45–500 mass range with a frequency of 8 spectra/s. Compounds were identified according to two orthogonal criteria: spectral similarity score (>800) and retention index (±20). After identification and peak area integration, compound quantification was carried out using the internal standards (ribitol, 5TMS and C17:0, 1TMS) and response factors obtained from the analysis of authentic standards (Appendix Tab. S5). When standards were unavailable, response factors from structurally-related molecules were used.

## Tissue permeability assay

*P. patens* tissue permeability was probed by immersing gametophores for 30 s in a 0.05% toluidine blue solution containing 2% tween20. Care was taken to remove rhizoids prior to staining assay and not to immerse the cut area. Gametophores were then abundantly rinsed with distilled water. For each genotype or replicate, ten gametophores were individually processed and subsequently pooled. Sample pigments were extracted in 400 μl of buffer (200 mM Tris-HCl, 250 mM NaCl, 25 mM EDTA) with two 3 mm steel balls operated at 30 Hz for 5 min. Following addition of 800 μl ethanol, samples were vortexed and centrifuged for 15 min at $18,000 \times g$. Absorbance of supernatants at 626 nm (A626) and 430 nm (A430) were recorded; A626/A430 ratio was used to quantify toluidine blue levels in plant tissue.

## Gene expression analysis by RT-qPCR

Total RNA was extracted from 5-week-old, lyophilized *P. patens* gametophores and *M. polymorpha* thalli using TriReagent (Sigma) and was subsequently treated with RQ1 DNAse (Promega). 150 ng (*P. patens*) or 1 μg (*M. polymorpha*) of DNase-treated total RNA was retro-transcribed with the SuperScript IV enzyme (ThermoFisher Scientific) and an oligo(dT)18, following the manufacturers' instructions. qPCR was performed in a 384-well plate, in a reaction volume of 10 μL containing 1 μl (*P. patens*) or 0.2 μl (*M. polymorpha*) RT reaction, 0.5 μM of each primer, and 5 μL of Master mix SYBR Green I (Roche). Reactions were run in triplicates using a LightCycler® 480 II (Roche) with the following program: 95 °C for 10 min, followed by 40 amplification cycles [95 °C for 10 s (denaturation) – 60 °C for 15 s (hybridization) – 72 °C for 15 s (extension)], and a melting curve analysis from 55 to 95 °C to verify primer specificity. Crossing points (Cp) were determined using manufacturer's software and corrected with primer pair efficiency. The expression levels of *P. patens* CYP73 genes were normalized to the expression levels of *PpTRX1* (*Pp3c19_1800*) and *PpSEC15* (*Pp3c27_3270*); the expression levels of *M. polymorpha* CYP73 genes were normalized to the expression levels of *MpACT7* (*Mp6g11010*) and *MpEF1* (*Mp3g2340*).

## *M. polymorpha* CRISPR/Cas9-mediated genome editing

Mutant alleles of *M. polymorpha* CYP73A1 were generated by CRISPR/Cas9 following the procedures described in Sugano et al (2018). Briefly, two independent protospacer sequences in CYP73A1 first exon were identified using the CRISPOR tool (Concordet and Haeussler, 2018). Selected protospacers started with a G for proper U6 promoter-driven expression and had a specificity score of 100. Double-stranded protospacer fragments were reconstituted by hybridization of two complementary oligonucleotides and were *Bsa*I-cloned into the Gateway pMpGE-En03 vector that contains Mp*U6-1* promoter and gRNA scaffold (Sugano et al, 2018). MpU6pro-gRNA expression cassettes were subsequently transferred by LR recombination into the binary pMpGE011 vector, which contains a *SpCas9* expression cassette and allows chlorsulfuron-based selection of plant transformants (Sugano et al, 2018). Recombined pMpGE011 vectors were introduced into *A. tumefaciens* GV3101 strain and served for the transformation of *M. polymorpha* Tak-1 accession according to the thallus method (Kubota et al, 2013). Transformants were selected on half-strength Gamborg B5 medium supplemented with 0.5 μM chlorsulfuron. Genome editing was checked in transformants by Sanger sequencing of PCR amplicons, using primers designed by the CRISPOR tool. Edited lines were further propagated from a single gemma to avoid mosaicism and confirm mutant allele.

## Piperonylic acid treatment

Standard agar and liquid growth media were supplemented with 50 μM piperonylic acid (PA) using 100 mM stock solution prepared in DMSO. Mock medium was prepared by supplementing growth media with 0.05% DMSO. PA treatment was initiated by transferring wild-type *C. richardii* sporophytes, *P. patens* gametophores, *M. polymorpha* gemmae and *A. agrestis* thallus pieces in PA and mock media. PA treatment of *K. nitens* was initiated by inoculating 180 mL PA-supplemented liquid culture with 20 mL standard pre-culture (1:10 final ratio).

## Statistical analysis

All statistical analyses were performed with the GraphPad v10 software. Unless otherwise stated, mean comparisons of wild type vs mutants or control vs treatment were performed using unpaired Student *t*-tests with *P*-values correction for multiple comparisons according to the Holm-Šídák method. Data for *M. polymorpha* thallus area and gemmae cup number were analyzed via one-way ANOVA with *P*-values correction for multiple comparisons according to the Dunnett method.

## Data availability

All the data produced in this study are available in supplementary materials as Source Data and Dataset files. Materials described in this study are available upon request. This study includes no data deposited in external repositories.

The source data of this paper are collected in the following database record: biostudies:S-SCDT-10_1038-S44318-024-00181-7.

## Peer review information

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

## Acknowledgements

HR received support from the initiative of excellence IDEX Unistra (ANR-10-IDEX-0002-02) and the Agence Nationale de la Recherche (ANR-19-CE20-0017). HR is grateful to the "Région Grand Est - Fonds Régional de Coopération pour la Recherche 2019" (VitEst project) for co-funding the GC-TOFMS equipment. SK and LK were supported by PhD fellowships from the Ministère de l'Enseignement supérieur et de la Recherche. KT was supported by a postdoctoral fellowship from the Japan Society for the Promotion of Science. RR acknowledges support from the German Research Foundation DFG under Germany's Excellence Strategy (EXC-2189, CIBSS). HR and RR acknowledge the support of the Freiburg Institute of Advanced Studies FRIAS and the University of Strasbourg Institute of Advanced Study USIAS to the METABEVO project. *Anthoceros agrestis* Bonn and *Ceratopteris richardii* Hn.n strains were kindly provided by Péter Szövényi (University of Zurich) and Andrew Plackett (University of Birmingham), respectively. We acknowledge the IBMP Gene Expression Analysis, Plant Imaging and Mass Spectrometry and Plant Production core facilities for their technical assistance.

## Author contributions

**Samuel Knosp**: Conceptualization; Formal analysis; Investigation; Writing—review and editing. **Lucie Kriegshauser**: Investigation; Writing—review and editing. **Kanade Tatsumi**: Formal analysis; Investigation; Writing—review and editing. **Ludivine Malherbe**: Investigation. **Mathieu Erhardt**: Investigation. **Gertrud Wiedemann**: Formal analysis; Investigation; Writing—review and editing. **Bénédicte Bakan**: Formal analysis; Investigation; Writing—review and editing. **Takayuki Kohchi**: Supervision; Funding acquisition; Methodology; Writing—review and editing. **Ralf Reski**: Supervision; Funding acquisition; Methodology; Writing—review and editing. **Hugues Renault**: Conceptualization; Data curation; Formal analysis; Supervision; Funding acquisition; Investigation; Visualization; Writing—original draft; Project administration.

Source data underlying figure panels in this paper may have individual authorship assigned. Where available, figure panel/source data authorship is listed in the following database record: biostudies:S-SCDT-10_1038-S44318-024-00181-7.

## Disclosure and competing interests statement

The authors declare no competing interests.

# Expanded View Figures

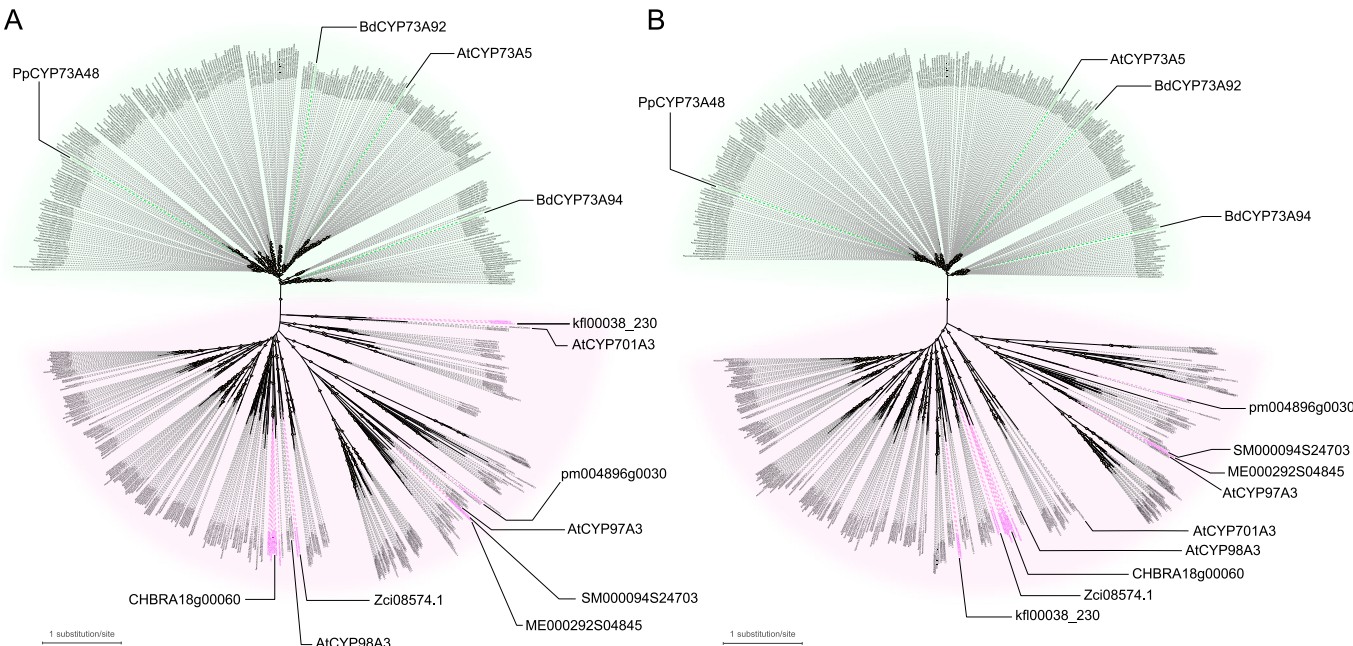

**Figure EV1.   Phylogenetic analysis of CYP73 and clan 71 CYP sequences.**

Maximum-likelihood trees of 275 CYP73 sequences and 271 additional outgroup sequences derived from *A. thaliana*, *P. patens* and charophytes (AtCYP73A5 tBLASTn bit-score >100). (**A**) Unrooted ML nucleotide tree (IQ-TREE2, SYM + I + R10). (**B**) Unrooted ML amino acid tree (IQ-TREE2, Q.insect+I + R8). Relevant CYP73 homologs, *A. thaliana* clan 71 CYP members and charophyte sequences corresponding to BLASTp best hits (see Appendix Tab. S1) are shown. Ultrafast bootstrap support values equal or superior to 80 are annotated on branch as yellow dots. Trees are drawn to scale; scale bars represent the number of substitutions per site.

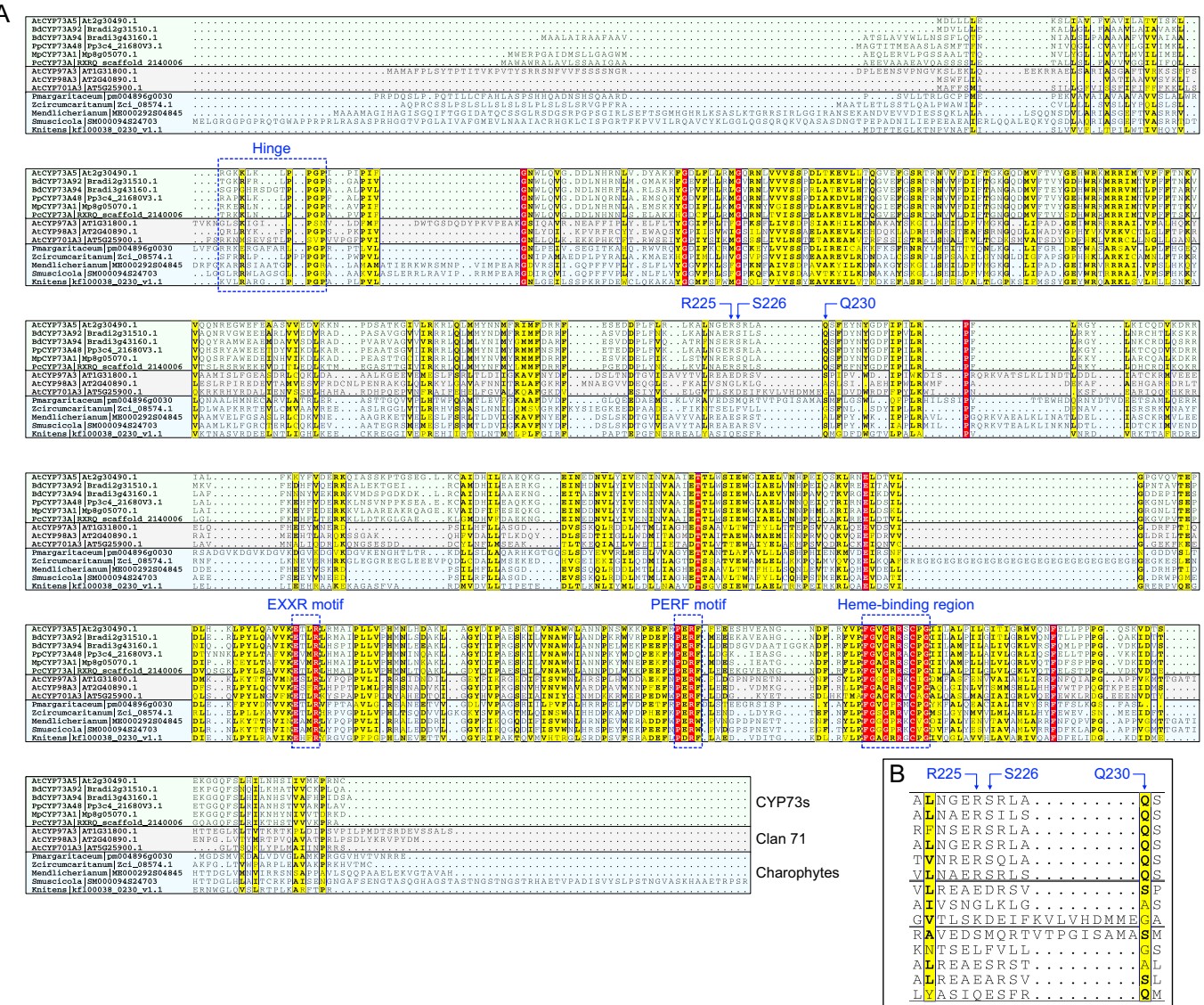

**Figure EV2.  Multiple sequence alignment of CYP73 proteins with clan 71 CYP members.**

(**A**) Full-length protein sequences of biochemically characterized CYP73 proteins from both tracheophytes (AtCYP73A5, BdCYP73A92, BdCYP73A94) and bryophytes (PpCYP73A48, MpCYP73A1, PcCYP73A) were aligned with *A. thaliana* clan 71 CYP members (AtCYP97A3, AtCYP98A3, AtCYP701A3). Protein sequence corresponding to BLAST best hits in the charophytes *P. margaritaceum*, *Z. circumcaritanum*, *M. endlicherianum*, and *K. nitens* (Fig. 1B) were included in the alignment. These charophyte proteins were found closely associated to *A. thaliana* clan 71 CYP members (Appendix Tab. S1, Fig. EV1). Alignment was performed with MUSCLE and visualized using ESPript 3.0. Positions that are identical are highlighted with a red background; positions with >70% similarity are highlighted with a yellow background. Cytochrome P450 conserved regions are highlighted. (**B**) Focus on the protein region encompassing the three residues critical for *t*-cinnamic acid binding in CYP73 active site.

