## [Peer Review File · The EMBO Journal]

An ancient role for CYP73 monooxygenases in phenylpropanoid biosynthesis and embryophyte development

Samuel Knosp, Lucie Kriegshauser, Kanade Tatsumi, Ludivine Malherbe, Mathieu Erhardt, Gertrud Wiedemann, Bénédicte Bakan, Takayuki Kohchi, Ralf Reski, and Hugues Renault

Corresponding author: Hugues Renault (renault@unistra.fr)

Review Timeline:

Submission Date:	29th Oct 23
Editorial Decision:	28th Nov 23
Revision Received:	1st May 24
Editorial Decision:	14th Jun 24
Revision Received:	3rd Jul 24
Accepted:	15th Jul 24

Editor: Yehu Moran

Transaction Report:

Dear Dr. Renault,

Thank you for submitting your manuscript for consideration by the EMBO Journal. It has now been seen by three referees whose comments are shown below.

Given the referees' positive recommendations, I would like to invite you to submit a revised version of the manuscript, addressing the comments of all three reviewers. I should add that it is EMBO Journal policy to allow only a single round of revision, and acceptance of your manuscript will therefore depend on the completeness of your responses in this revised version.

Please give special attention to the comments of Reviewer #1 regarding the phylogeny and the outgroup you used. These seem to be very important for supporting (or refuting) some of the evolutionary claims made in the paper and we expect you to address this issue thoroughly and perform the analysis according to their suggestions. Should you have any reservations, please explain them in detail in your response letter. Moreover, we encourage you to explore their idea of expanding your experiments in additional bryophyte species. Additionally, the idea by Reviewer #3 to expand the PA treatment to additional species of ferns and/or algae to expand our evolutionary understanding on this topic seems to us worthwhile. Should these suggestions for additional experiments be too technically challenging, we would ask you to explain this issue in detail in your response letter. Please bear in mind that we may also further consult with experts in the field regarding the feasibility of such experiments in light of your explanations.

Thank you for the opportunity to consider your work for publication. I look forward to your revision.

Kind regards,

Yehu Moran
Academic Editor
The EMBO Journal

- a point-by-point response to the referees' comments, with a detailed description of the changes made (as a word file).
- a word file of the manuscript text.
- individual production quality figure files (one file per figure)
- a complete author checklist, which you can download from our author guidelines (<https://www.embopress.org/page/journal/14602075/authorguide>).

- Expanded View files (replacing Supplementary Information)

We realize that it is difficult to revise to a specific deadline. In the interest of protecting the conceptual advance provided by the work, we recommend a revision within 3 months (26th Feb 2024). Please discuss the revision progress ahead of this time with the editor if you require more time to complete the revisions.

Referee #1:

- general summary and opinion about the principal significance of the study, its questions and findings

The manuscript from Knosp et al. explores the ancient function of CYP73 monooxygenases in the development of embryophytes, focusing on cinnamic acid 4-hydroxylase (hereinafter referred as C4H). The study includes diverse approaches from evolutionary genomics and genetics to metabolomics and in planta evidence, all aimed at unraveling the evolution and function of the CYP73 gene family. What stands out is not just the diversity of methodologies used but also the rich evolutionary evidence drawn from mutants across various model plants to prove the evolutionary conservation of CYP73 's biological function. The novelty of this study may not be groundbreaking since 1) prior studies (including those from the same research group) have already explored related phenolic metabolisms with the plant terrestrialization aspect [e.g., PMID: 28270693] and previous work on CYP73 orthologs in land plants showed similar functions in bryophytes [e.g., PMID: 19682296]. However, the strength lies in the variety of methodologies and species they used, which strengthens the robustness of their findings. The study not only confirms the known role of C4H in flowering plants, but it also extends this evidence to a wide range of bryophytes, providing a unique perspective on C4H function, developmental physiology, and plant terrestrialization evolution.

Despite some concerns, I am in favor of the manuscript's publication in this journal due to its multidisciplinary approach and its contribution to key evolutionary aspects of embryophytes. While the experiments and conclusions are generally robust, addressing essential issues in a revision would further enhance the manuscript's overall quality.

##

- specific major concerns essential to be addressed to support the conclusions

While I appreciate the manuscript's strengths in presenting a diverse set of results to support its claims, I can't help but notice the absence of an outgroup and the skewed perspective of the ingroup in Figures 1 & 2. This raises concerns about the accuracy of the evolutionary interpretation. To enhance the paper's reliability and depth in evolutionary analysis, I suggest incorporating additional analyses and references to substantiate the proposed claims. Addressing these issues would strengthen the overall credibility of the manuscript.

Missing a comparable group and a distorted view of the CYP73 genes, as shown in Figures 1 and 2, raises concerns about some of their evolutionary interpretations. Because the manuscript isn't reporting a completely new function of this enzyme but rather demonstrating its evolutionary conservation of C4H. Despite outgroup control being required for evolutionary interpretation to explain CYP73 specificity, the manuscript simply describes conserved motifs without comparing them to others.

The manuscript claims that CYP73 appeared suddenly in early plants and contributed to the plant tertiarization process. I agree with the importance of the C4H function and the role they claim C4H might play, but there are many logical gaps in the emergence of CYP73 that they did not address in the manuscript. Several papers have already completed a comprehensive survey of CYP in plants, revealing a common ancestor with CYP73 and providing potential candidates for comparison with CYP73 proteins (hereafter, outgroup) [PMID: 15208422, 21443632, 22303269, 34216829]. The origin of the CYP73 gene is not stated specifically in the manuscript, but it is clear from the references that there are potential candidates. According to a

reference study of plant CYP evolution, it seems like the CYP73 gene is the member classified as "Clan 71". Some reference papers identified the most closely related CYP family of CYP73 as the CYP98 gene family [PMID: 11429408, 15208422], which was also discovered in Table S1. But I don't have enough background, so I guess for sure the authors can do the phylogenetic analysis of "Clan 71" to define the ancestral origin of CYP73. It is not necessary to include all of "Clan 71" in their phylogenetic analysis, however, they must use a reasonable outgroup. There are still remaining questions, for example,

- What distinguishes CYP73 from other members of the cytochrome P450 (CYP) enzyme family (e.g., Clan 71)?
- Is there a specific combination of domains that results in significant variations, or is it primarily differences in motifs that have led to the diversification of the CYP73 gene family?
- What is the domain composition of the CYP73 gene, and are there differences within the CYP73 gene family?

This is particularly significant as the paper discusses the evolutionary emergence of novel functionality in CYP73 gene family. To clarify and support their claim, they must show that other CYP73 genes do not have this specific motif and that it is unique to CYP73. Figure 2B, for example, can be shown in conjunction with what I would suggest showing motif alignment with other "Clan 71" members (Figure 2B).

This has been briefly discussed in Figure 1B and Table S1, but the details are not sufficient for their claim and should be revised. This should be improved by stating more details at the start or conducting additional analysis. For example, I understand the purpose of presenting Figure 1B, but I don't think it is clearly supporting the manuscript's argument. Because BLASTp results may reflect protein homology because of evolutionary divergence, this does not represent CYP73 evolution. For Figure 1B, I'd like to suggest a refinement, perhaps removing the 'forward BLASTp' and 'RBH' categories and instead focusing on illustrating the presence of core motifs (e.g., R226, Q230) or highlighting specific domains of CYP73 could strengthen its relevance. Furthermore, demonstrating the presence & absence or copy numbers after a new phylogenetic analysis of "Clan 71" will provide a more robust justification for the phylogenetic analysis.

##

##

- minor concerns that should be addressed

(1) Phylogenetic analysis and tree: Fig. 1C

I noticed that in Figure 1C, the authors used midpoint rooting due to the absence of an outgroup. As I mentioned previously, I'm not sure about the reasoning behind this choice and it should be revised. Aside from that, I noticed in Data S2 that the midpoint rooting result is "62.1/80", but this supporting value is missing from Figure 1C. Furthermore, there appears to be some difficulty when looking at the bootstrap values in Figure 1C, and the major group supporting values do not appear to align with the original tree in Data S2. For example, the ancestral branch of hornwort is "76.8/87" in the original tree, but "77.8/87" in Figure 1C. I haven't gone over everything, but please check these discrepancies.

The authors used codon-based nucleotide alignment to build the dataset for the phylogenetic analysis in this study. The pairwise identity of this alignment, which is around 66%, may not be the best fit for constructing a phylogeny. They can still use the previous method, but considering the evolutionary distance of taxa, a protein alignment-based phylogenetic tree seems to be more appropriate.

(2) Broader consideration of motif conservation

I understand the focus on the catalytic binding motifs, as they are undoubtedly important for the protein's function based on the given results. The choice to create mutants within this highly conserved motif aligns well with the protein's overall structure. However, the transition to catalytic motifs in post-phylogenetic analysis felt a bit abrupt to me. To enrich the depth of your analysis, I'd suggest delving into the conservation details of other amino acid residues in the CYP73 protein. It could be beneficial to integrate information from residue conservation scores derived through alignments, offering insights into how the conservation patterns of structurally or functionally significant residues compare to others.

(3) Difference between paralogs in each species and expression profiles of bryophytes: Fig. 4A, 5A

I feel there's a bit of a gap in the manuscript when it comes to the description of CYP73 in each species. It might be beneficial to delve a bit more into the details beyond just the expression-level differences. I'd suggest a brief exploration of the different proteins, perhaps by visualizing their sequence differences. I think it would be really insightful if the authors could delve a bit deeper into each paralog. What's causing the differences, maybe something in the promoter, the length, specific domains, or motifs? A bit more detail on these aspects could add valuable depth. Also, taking a reference from the previous work and using a similar visualization method might be a great way to illustrate these points effectively [PMID: 28505373].

I have a concern with the expression levels of the paralogs, particularly the strong upregulation of PpCYP73A49 in Physcomitrium's sporophyte tissue. This prompts the thought that similar distinctions might be present in other species like Marchantia. It could be worthwhile to include Marchantia sporophyte expression data in Figure 5A, given its availability. I understand the challenges with genetic manipulation in Anthoceros but including its expression levels from the reference paper [PMID: 32170292] could provide a more comprehensive picture. I also noticed in the figure descriptions, for example in Fig. 3A, the source of the transcriptome data isn't clearly mentioned. It would be helpful to specify and refer to the exact data source in

the methods section for clarity.

(4) Precision in graphical representation of plots: 'HPLC_UV chromatogram plot'

I noticed some issues with the way chromatogram plots are presented, particularly in Figures 3C, 5F, and 4A. The axis values are not clearly indicated, and in the case of Fig. 4A, there seem to be incorrect y-axis values. It would greatly improve clarity if you could revise these plots or, at the very least, include auxiliary lines to guide readers.

(5) Permeability assay

I noticed the permeability assay in Fig. 4G, and it got me thinking about its potential application in other bryophytes. It seems like a relatively feasible addition, and I'd recommend extending this experiment to additional species and mutants, which could provide stronger evidence for the defects in cuticle biogenesis. Additionally, considering the treatment with piperonylic acid in Figure 6A, incorporating a permeability assay could further enrich the findings and provide a more comprehensive understanding of the experimental outcomes.

(6) Piperonylic acid

I couldn't find clear evidence of piperonylic acid's actual binding site. If known, it would be great to have a clear statement. Also, considering the potential triggering of other effects with PA treatment, it might be insightful to include a control in the study. Treating PA to the mutants that have lost sensitivity to piperonylic acid could help demonstrate that the observed effects are indeed related to the presence of PA in the wild type.

(7) Minor details

- Line 41, 43, 100 "Viridiplantae" should not be italicized.
- Line 215 "Fig. 4D": isn't this Fig. 4G?
- Table S1: Line 949 'A. thaliana' and Line 951 'Arabidopsis thaliana' should be italicized.

##

##

- any additional non-essential suggestions for improving the study (which will be at the author's/editor's discretion)

- I don't have enough background to comment, but the introduction is too general and vague, and it also doesn't really address land plant evolution in relation to terrestrialization. I would suggest adding general plant evolution related to terrestrialization and linking this to C4H evolution.
- The sentence from Line 215-217, "To achieve... M. polymorpha." comes across as vague and somewhat confusing. I would suggest considering its removal for clarity and coherence in the manuscript.
- Regarding the phylogenetic analysis, it's noted that the ultrafast bootstrapping ('-B' option in IQ-Tree) was employed for bootstrapping, which involves pseudoreplicates. If the authors choose to reconstruct the tree with an outgroup, I recommend considering the use of the standard nonparametric bootstrapping method ('-b' option) in addition to the ultrafast bootstrapping method. However, this suggestion is optional and depends on the authors' discretion.
- The key term "terrestrialization" is far too broad to be mentioned in this manuscript. I would suggest changing the key word to something more specific (e.g., CYP73 evolution, cuticle, C4H).
- I don't understand the Figure S2 result. Could you please explain how to interpret it in greater detail?

##

Referee #2:

In this manuscript, the authors have undertaken a series of experiments to characterize the C4H enzyme in bryophytes, aiming to shed light on the evolution of land plants linked to the enzymatic activity of phenylpropanoid biosynthesis. They present evidence suggesting that the origin of C4H can likely be traced back to the common ancestor of land plants. Additionally, they report that C4H in algae was undetectable using their BLAST-based method, aligning with previous studies. In vitro assays were successful in confirming the conserved activity of C4H in bryophyte homologs, including the identification of a specific residue crucial for C4H activity. The study further examines knock-out phenotypes in *Physcomitrium patens* and *Marchantia polymorpha*, uncovering a conserved role in cuticle formation and highlighting its significance in development. The research is methodologically sound, albeit with few unexpected findings, and it significantly contributes to our understanding of the evolutionary conservation of the important enzyme in land plants.

The manuscript is generally well-written and provides an engaging read. However, there are specific areas where clarification or additional information would enhance the reader's understanding:

L106: The term 'homology' denotes shared ancestry. In the context of BLAST hits, the use of 'sequence similarity/identity' would be a more accurate terminology than 'homology'.

L113: The manuscript details the presence of multiple CYP73 paralogs in bryophytes but selects only one per species for

functional analysis without initial justification. Although this might be explained later in the text, a brief rationale at this point would enhance reader understanding. Clarification on the selection criteria among these paralogs is recommended.

L165: The description of *uidA* reporter lines is somewhat cursory. It would be beneficial to specify whether these genes are expressed in the epidermis or the inner tissues of gametophyte stems. This detail is pertinent to the discussion on 'waterproofing' capabilities introduced earlier and is also relevant to the subsequent analysis of cuticular composition and function.

L189: The phrase 'complete "CYP73" deficiency' seems vague. Given the existence of untargeted CYP73 paralogs, a more precise description ("C4H"?) of the deficiency would be beneficial.

L286: The term "evolution reconstruction" is a bit ambiguous. If it refers to 'phylogenetic analysis', using this specific term would prevent potential misinterpretation.

L338: The observation that exogenous p-coumaric acid did not fully rescue the phenotype suggests a considerable likelihood that the remaining unrescued phenotypes may be attributable to the hyperaccumulation of t-cinnamic acid.

The initial paragraph of the Discussion section could potentially be more fitting in the Introduction, as it primarily contextualizes the research without delving into specific results.

Fig. 4A x-axis label: It appears that "Time" would be a more appropriate label than "Temps".

Referee #3:

Knosp et al. provide new data on the functional origin of C4H enzymes in plants. Beginning with phylogenetic analysis of C4H enzymes, the authors identify and then functionally validate C4H in bryophytes through a combination of molecular genetics, biochemistry, and analytical techniques. This focused examination of phenylpropanoid pathway evolution serves as an excellent example of how to leverage non-flowering/non-vascular systems to answer key questions in plant biology and biochemistry. I am largely supportive of the manuscript overall and have only a few concerns/comments.

1. Genetic complementation/rescue: The data would be strengthened by rescuing the mutants generated in *Physcomitrium* and *Marchantia*. Moreover, the authors could provide data on the characterization of *Arabidopsis* mutant rescues (for example, westerns or RT-PCR). This is especially useful for the PpCYP line that failed to rescue *At* mutants (i.e. is this due to low transgene expression?)

2. Is it possible to assess cuticle integrity across all bryophytes? Given the analytical data, it could be informative to assess whether the *Marchantia* mutants are similarly impacted through toluidine blue assays. This could be further extended to the piperonylic acid treatments if feasible.

3. An intriguing implication of the data is that streptophyte algae lacking C4H would be insensitive to PA treatment. Providing a wider exploration C4H inactivation via PA (algae, ferns, etc) could strengthen the evolutionary assertions mentioned throughout the manuscript. I appreciate that these systems are not trivial to set-up, so this is only a suggestion provided that the assays/experiments were feasible for the authors.

4. If C4H acts early in the phenylpropanoid pathway, then how do other phenylpropanoids accumulate in *Marchantia* mutants? Is there a genetic redundancy issue here, as in *Physcomitrium*? Clarity on this could be useful, perhaps in the discussion.

Response to Referees

We are grateful to the three reviewers for sharing their time and expertise in evaluating our manuscript. As you will see below, we have been able to address most of the comments regarding CYP73 evolution/phylogenetic analysis and have performed additional experiments with two new models, a fern and a charophyte, to strengthen our conclusions as it has been suggested.

The following new data are disclosed in the revised manuscript:

- **Figure EV1:** new phylogenetic analysis comprising 275 CYP73 sequences and 271 additional outgroup sequences from *P. patens*, *A. thaliana* and charophytes.
- **Figure EV2:** new full length protein alignment including characterized CYP73 homologs along with Clan 71 and charophyte CYPs.
- **Figure 3E-F:** pictures of gametophore cross-sections from *uidA* reporter lines
- **Figure 5A:** *MpCYP73* expression data from sporophytes
- **Figure 6** has been profoundly altered. It now reports the impact of piperonylic acid treatment in five different species – *Ceratopteris richardii* (fern, tracheophyte), *Physcomitrium patens* (moss, bryophyte), *Marchantia polymorpha* (liverwort, bryophyte) and *Klebsormidium nitens* (Klebsormidiophyceae, charophyte) – on development, UV metabolic profiles and level of shikimate phenolic esters.

Below are point-by-point answers to reviewers' comments.

Sincerely yours,

Hugues Renault, corresponding author.

Referee #1:

- general summary and opinion about the principal significance of the study, its questions and findings

The manuscript from Knosp et al. explores the ancient function of CYP73 monooxygenases in the development of embryophytes, focusing on cinnamic acid 4-hydroxylase (hereinafter referred as C4H). The study includes diverse approaches from evolutionary genomics and genetics to metabolomics and in planta evidence, all aimed at unraveling the evolution and function of the CYP73 gene family. What stands out is not just the diversity of methodologies used but also the rich evolutionary evidence drawn from mutants across various model plants to prove the evolutionary conservation of CYP73's biological function. The novelty of this study may not be groundbreaking since 1) prior studies (including those from the same research group) have already explored related phenolic metabolisms with the plant terrestrialization aspect [e.g., PMID: 28270693] and previous work on CYP73 orthologs in land plants showed similar functions in bryophytes [e.g., PMID: 19682296]. However, the strength lies in the variety of methodologies and species they used, which strengthens the robustness of their findings. The study not only confirms the known role of C4H in flowering plants, but it also extends this evidence to a wide range of bryophytes, providing a unique perspective on C4H function, developmental physiology, and plant terrestrialization evolution.

Despite some concerns, I am in favor of the manuscript's publication in this journal due to its multidisciplinary approach and its contribution to key evolutionary aspects of embryophytes. While the experiments and conclusions are generally robust, addressing essential issues in a revision would further enhance the manuscript's overall quality.

- specific major concerns essential to be addressed to support the conclusions

While I appreciate the manuscript's strengths in presenting a diverse set of results to support its claims, I can't help but notice the absence of an outgroup and the skewed perspective of the ingroup in Figures 1 & 2. This raises concerns about the accuracy of the evolutionary interpretation. To enhance the paper's reliability and depth in evolutionary analysis, I suggest incorporating additional analyses and references to substantiate the proposed claims. Addressing these issues would strengthen the overall credibility of the manuscript.

Missing a comparable group and a distorted view of the CYP73 genes, as shown in Figures 1 and 2, raises concerns about some of their evolutionary interpretations. Because the manuscript isn't reporting a completely new function of this enzyme but rather demonstrating its evolutionary conservation of C4H. Despite outgroup control being required for evolutionary interpretation to explain CYP73 specificity, the manuscript simply describes conserved motifs without comparing them to others.

The manuscript claims that CYP73 appeared suddenly in early plants and contributed to the plant tertiarization process. I agree with the importance of the C4H function and the role they claim C4H might play, but there are many logical gaps in the emergence of CYP73 that they did not address in the manuscript. Several papers have already completed a comprehensive survey of CYP in plants, revealing a common ancestor with CYP73 and providing potential candidates for comparison with CYP73 proteins (hereafter, outgroup) [PMID: 15208422, 21443632, 22303269, 34216829]. The origin of the CYP73 gene is not stated specifically in the manuscript, but it is clear from the references that there are potential candidates. According to a reference study of plant CYP evolution, it seems like the CYP73 gene is the member classified as "Clan 71". Some reference papers identified the most closely related CYP family of CYP73 as the CYP98 gene family [PMID: 11429408, 15208422], which was also discovered in Table S1. But I don't have enough background, so I guess for sure the authors can do the phylogenetic analysis of "Clan 71" to define the ancestral origin of CYP73. It is not necessary to include all of "Clan 71" in their phylogenetic analysis, however, they must use a reasonable outgroup. There are still remaining questions, for example,

Response 01: Following reviewer 1 recommendation, we have extended our CYP73 phylogeny (**Fig. 1C**) with 271 additional sequences from *P. patens*, *A. thaliana* and charophytes (CYP73A5-BLAST bit-score >100) (**Fig. EV1**; see below) to serve as outgroups. Both nucleotide (panel A) and amino acid (panel B) phylogenies show that CYP73 sequences group within a well-defined monophyletic group. New phylogenies further demonstrate that charophyte sequences are closer to other clan 71 members, such as CYP97, CYP701 or CYP98. These findings align very well with our initial reciprocal best hit (RBH) survey (**Fig. 1B, Appendix Tab. S1**). We therefore decided on keeping the RBH data in **Fig. 1** as they remain valid and are easier to understand at a glance; they are now backed by the new phylogenetic analyses provided in **Fig. EV1**.

In addition, our new phylogenies support an origin of the CYP73 family within the CYP71 clan, as it has been proposed in earlier studies. However, they fail to pinpoint the precise origin or a clear sister family. This still leaves the ancestral origin of the CYP73 family unresolved, akin previous attempts (e.g. PMID: 34165823, 2850537). Upcoming genomic data, especially from charophytes, may help solve this mystery in the future.

Text has been amended accordingly (**L103-109**).

Figure EV1. Phylogenetic analysis of CYP73 and clan 71 CYP sequences.

Maximum-likelihood (ML) trees of 275 CYP73 sequences and 271 additional sequences derived from *A. thaliana*, *P. patens* and charophytes as outgroups (AtCYP73A5 tblastn bit-score > 100). (A) Unrooted ML nucleotide tree (IQ-TREE2, SYM+I+R10). (B) Unrooted ML amino acid tree (IQ-TREE2, Q.insect+I+R8). Representative CYP73 homologs, *A. thaliana* clan 71 CYP members and charophyte sequences corresponding to BLASTp best hits (see Appendix Tab. S1) are shown. Ultrafast bootstrap support values equal or superior to 80 are annotated on branch as yellow dots. Trees are drawn to scale; scale bars represent the number of substitutions per site.

- What distinguishes CYP73 from other members of the cytochrome P450 (CYP) enzyme family (e.g., Clan 71)?

Response 02: Cytochrome P450 enzymes are ubiquitous proteins spanning all life kingdoms. They overall share low sequence identity but high structural similarity. Unlike to some other protein families (e.g. Rho proteins, PMID: 38039969), CYP families cannot be defined by specific domains, indeed their respective functions are underpinned by discrete sequence changes scattered along the whole protein (apart from a few highly conserved core domains). It is worth noting that the whole cytochrome P450 diversity, from procaryotes to eukaryotes, is associated to a unique Pfam domain (i.e., PF00067).

- Is there a specific combination of domains that results in significant variations, or is it primarily differences in motifs that have led to the diversification of the CYP73 gene family?

Response 03: CYP functions relying on discrete sequence variations that do not form domains *per se*, a domain combination can thus be excluded as a determinant of CYP73 diversity (see **response 02**). Instead, we reported in our manuscript (L134-L161) three residues in the F helix that support the canonical C4H activity. These residues are critical for CYP73 function and likely accompanied the emergence and evolution of this CYP family. Subsequent diversification of the CYP73 family most likely results from adaptation to various cellular environments, including membrane and protein interactions. For instance, we previously reported that two major CYP73 classes evolved in seed plants (PMID: 2850537). Although both classes efficiently catalyzed C4H activity, they greatly differed in their N-terminal transmembrane domains, leading to differential membrane topology. Apart from the N-terminal transmembrane domain, no class-defining domains were evident from sequence analysis.

- What is the domain composition of the CYP73 gene, and are there differences within the CYP73 gene family?

This is particularly significant as the paper discusses the evolutionary emergence of novel functionality in CYP73 gene family. To clarify and support their claim, they must show that other CYP73 genes do not have this specific motif and that it is unique to CYP73. Figure 2B, for example, can be shown in conjunction with what I would suggest showing motif alignment with other "Clan 71" members (Figure 2B).

This has been briefly discussed in Figure 1B and Table S1, but the details are not sufficient for their claim and should be revised. This should be improved by stating more details at the start or conducting additional analysis. For example, I understand the purpose of presenting Figure 1B, but I don't think it is clearly supporting the manuscript's argument. Because BLASTp results may reflect protein homology because of evolutionary divergence, this does not represent CYP73 evolution. For Figure 1B, I'd like to suggest a refinement, perhaps removing the 'forward BLASTp' and 'RBH' categories and instead focusing on illustrating the presence of core motifs (e.g., R226, Q230) or highlighting specific domains of CYP73 could strengthen its relevance. Furthermore, demonstrating the presence & absence or copy numbers after a new phylogenetic analysis of "Clan 71" will provide a more robust justification for the phylogenetic analysis.

Response 04: Sequence variation within the CYP73 family does exist, evident for instance from the 71% identity shared by moss CYP73A48 and Arabidopsis CYP73A5 proteins. However, as detailed in **responses 02 and 03**, we could not identify a CYP73-specific domain as sequence variations are rather discrete.

We now provide a multiple sequence alignment including biochemically characterized CYP73 protein sequences (from both tracheophytes and bryophytes), clan 71 members (CYP97, CYP98, CYP701) and sequences corresponding to best hits in charophytes (**Fig. EV2**, see below). This alignment shows that the R225, S226 and Q230 residues are found only in CYP73 proteins (**Fig. EV2B**). Text has been amended accordingly (**L145-149**).

As detailed in **response 01**, we have also conducted new phylogenetic analyses that include outgroup sequences (**Fig. EV1**). New phylogenies are in accordance with RBH results, showing that the CYP73 family is restricted to embryophytes. We think that **Fig. 1B**, **Fig. EV1** and **Fig. EV2** collectively provide compelling evidence in support of an emergence of the CYP73 family in an ancestor of embryophytes.

A

Figure EV2. Multiple sequence alignment of CYP73 proteins with clan 71 CYP members.

(A) Full-length protein sequences of biochemically characterized CYP73 proteins from both tracheophytes (AtCYP73A5, BcCYP73A92, BcCYP73A94) and bryophytes (PpCYP73A48, MpCYP73A1, PcCYP73A) were aligned with *A. thaliana* clan 71 CYP members (AtCYP97A3, AtCYP98A3, AtCYP701A3). Protein sequence corresponding to blast best hits in the charophyte *P. margaritaceum*, *Z. circumcaritanum*, *M. endlicherianum* and *K. nitens* (Fig. 1B) were included in the alignment. These charophyte proteins were found closely associated to *A. thaliana* clan 71 CYP members (Appendix Tab. S1, Fig. EV1). Alignment was performed with MUSCLE and visualized using the ESPrnt 3.0. Positions that are identical are highlighted with a red background; positions with >70% similarity are highlighted with a yellow background. Cytochrome P450 conserved regions are highlighted. (B) Focus on the protein region encompassing the three residues critical for *t*-cinnamic acid binding in CYP73 active site.

- minor concerns that should be addressed

(1) Phylogenetic analysis and tree: Fig. 1C

I noticed that in Figure 1C, the authors used midpoint rooting due to the absence of an outgroup. As I mentioned previously, I'm not sure about the reasoning behind this choice and it should be revised.

Response 05: The low sequence conservation between CYP73 and outgroup CYPs visible in **Fig. EV1** results in poor sequence alignment (for example, AtCYP73A5 closest homolog in *A. thaliana* shares only 33% identity), which in turn precludes the precise reconstruction of phylogenetic relationships within the CYP73 family. We therefore decided on keeping our initial, fine-grained phylogenetic analysis based solely on the 275 CYP73 sequences (**Fig. 1C**). To polarize the phylogenetic tree, and because an outgroup would have been detrimental to sequence alignment quality, we used the midroot approach. Resulting rooted tree topology was consistent with embryophyte phylogeny, giving confidence in this rooting approach.

Text has been amended accordingly (**L104-110** and **L116-117**).

Aside from that, I noticed in Data S2 that the midpoint rooting result is "62.1/80", but this supporting value is missing from Figure 1C. Furthermore, there appears to be some difficulty when looking at the bootstrap values in Figure 1C, and the major group supporting values do not appear to align with the original tree in Data S2. For example, the ancestral branch of hornwort is "76.8/87" in the original tree, but "77.8/87" in Figure 1C. I haven't gone over everything, but please check these discrepancies.

Response 06: Thank you for bringing this up! We have amended the tree support values in **Fig. 1C**.

The authors used codon-based nucleotide alignment to build the dataset for the phylogenetic analysis in this study. The pairwise identity of this alignment, which is around 66%, may not be the best fit for constructing a phylogeny. They can still use the previous method, but considering the evolutionary distance of taxa, a protein alignment-based phylogenetic tree seems to be more appropriate.

Response 07: We tested both protein- and nucleotide-based phylogenetic analyses. CYP73 nucleotide phylogenies proved more robust according to tree topology (recapitulates embryophyte systematics) and branch supports.

(2) Broader consideration of motif conservation

I understand the focus on the catalytic binding motifs, as they are undoubtedly important for the protein's function based on the given results. The choice to create mutants within this highly conserved motif aligns well with the protein's overall structure. However, the transition to catalytic motifs in post-phylogenetic analysis felt a bit abrupt to me. To enrich the depth of your analysis, I'd suggest delving into the conservation details of other amino acid residues in the CYP73 protein. It could be beneficial to integrate information from residue conservation scores derived through alignments, offering insights into how the conservation patterns of structurally or functionally significant residues compare to others.

Response 08: Thank you for the suggestion. Residue conservation scores could indeed bring interesting insights and possibly uncover important regions/motifs. However, we rather chose to follow a bottom-up approach relying on the identification of critical residues supporting C4H activity via homology modeling and docking experiments. Conservation of these particular residues was then investigated and visualized via a weblogo representation, confirming their high conservation in CYP73 proteins (**Fig. 2B**). This approach proved valid since the functional importance of the identified residues has been confirmed by in vitro assays. Newly created **Fig. EV2** now brings perspective on the conservation of such critical residues in other CYPs, where they were not found.

Text has been amended accordingly (**L145-149**).

(3) Difference between paralogs in each species and expression profiles of bryophytes: Fig. 4A, 5A

I feel there's a bit of a gap in the manuscript when it comes to the description of CYP73 in each species. It might be beneficial to delve a bit more into the details beyond just the expression-level differences. I'd suggest a brief exploration of the different proteins, perhaps by visualizing their sequence differences. I think it would be really insightful if the authors could delve a bit deeper into each paralog. What's causing

the differences, maybe something in the promoter, the length, specific domains, or motifs? A bit more detail on these aspects could add valuable depth. Also, taking a reference from the previous work and using a similar visualization method might be a great way to illustrate these points effectively [PMID: 28505373].

Response 09: Thank you for this comment. We think that the main strength of our manuscript is to provide experimental data on *CYP73* gene function, hence reaching beyond a descriptive approach on gene/protein structures. We consider that gathering such information from public databases would bring limited new functional insights and therefore decided not to follow this recommendation.

I have a concern with the expression levels of the paralogs, particularly the strong upregulation of PpCYP73A49 in *Physcomitrium*'s sporophyte tissue. This prompts the thought that similar distinctions might be present in other species like *Marchantia*. It could be worthwhile to include *Marchantia* sporophyte expression data in Figure 5A, given its availability. I understand the challenges with genetic manipulation in *Anthoceros* but including its expression levels from the reference paper [PMID: 32170292] could provide a more comprehensive picture. I also noticed in the figure descriptions, for example in Fig. 3A, the source of the transcriptome data isn't clearly mentioned. It would be helpful to specify and refer to the exact data source in the methods section for clarity.

Response 10: Following recommendation, we added *MpCYP73* sporophyte expression values to Fig. 5A.

Instead of a genetic approach targeting specific *CYP73* paralogs, we used a selective inhibitor to overall block C4H activity in *A. agrestis*. In this context, expression analysis of *A. agrestis* *CYP73* paralogs was not relevant to our study design, hence not included in the manuscript.

As to RNAseq expression data, we refer to the databases (CoNekT, MarolBase) from which expression values (TPM) have been retrieved. This information is visible in legends of Fig. 3A and Fig. 5A.

(4) Precision in graphical representation of plots: 'HPLC_UV chromatogram plot'

I noticed some issues with the way chromatogram plots are presented, particularly in Figures 3C, 5F, and 4A. The axis values are not clearly indicated, and in the case of Fig. 4A, there seem to be incorrect y-axis values. It would greatly improve clarity if you could revise these plots or, at the very least, include auxiliary lines to guide readers.

Response 11: After examination of chromatogram plots, we confirm that axes are properly labelled and to scale. Fig. 4A derives, with minor modifications, from the display report from Compass Data Analysis software (see figure below). Plot layouts of Fig. 4A, Fig. 3C and Fig. 5F in addition corresponds to standard visualizations of chromatogram data. We however agree that the continuous Y axis of Fig. 4A might be misleading. For better readability, we rescaled the axes to set the baseline to 0 for each sample. We have also slightly altered Fig. 3C and Fig. 5F to, hopefully, avoid any confusion.

(5) Permeability assay

I noticed the permeability assay in Fig. 4G, and it got me thinking about its potential application in other bryophytes. It seems like a relatively feasible addition, and I'd recommend extending this experiment to additional species and mutants, which could provide stronger evidence for the defects in cuticle biogenesis. Additionally, considering the treatment with piperonylic acid in Figure 6A, incorporating a permeability assay could further enrich the findings and provide a more comprehensive understanding of the experimental outcomes.

Response 12: This is a good point! In addition to toluidine blue assays in *P. patens* (Fig. 4F), we performed the assay in *M. polymorpha* and *A. agrestis* but encountered technical problems. Thalloid shape of both species made it very difficult to prevent rhizoids from massively sucking up toluidine blue during the assay (see picture A below). This led to rather unspecific and non-reproducible staining pattern which, in our view, couldn't be interpreted. Similar issues applied to the $\Delta PpCYP73A48/CYP73A49$ double mutants (briefly commented L228-229). Piperonylic acid treatment of wild-type *P. patens* phenocopies the $\Delta PpCYP73A48/CYP73A49$ double knock-out mutant and thus makes the interpretation of toluidine blue assays unreliable. Additional toluidine blue assays were therefore not performed.

(6) Piperonylic acid

I couldn't find clear evidence of piperonylic acid's actual binding site. If known, it would be great to have a clear statement. Also, considering the potential triggering of other effects with PA treatment, it might be insightful to include a control in the study. Treating PA to the mutants that have lost sensitivity to piperonylic acid could help demonstrate that the observed effects are indeed related to the presence of PA in the wild type.

Response 13: Binding site of piperonylic acid in CYP73 proteins is not known to the best of our knowledge. It has a carboxylic function; we can therefore speculate that it interacts with the same residues as *t*-cinnamic acid (R225). We did not treat moss CYP73 mutants with piperonylic acid as they already had severe developmental defects, which would have made interpretation rather difficult.

(7) Minor details

- Line 41, 43, 100 "Viridiplantae" should not be italicized.

Response 14: We have made the change, thank you!

- Line 215 "Fig. 4D": isn't this Fig. 4G?

Response 15: Line 215: "...despite the moderate compositional changes (Fig. 4D)." We confirm that we refer to the correct figure, Fig. 4D, relative to cutin composition.

- Table S1: Line 949 'A. thaliana' and Line 951 'Arabidopsis thaliana' should be italicized.

Response 16: We have made both changes, thank you!

- any additional non-essential suggestions for improving the study (which will be at the author's/editor's discretion)

- I don't have enough background to comment, but the introduction is too general and vague, and it also doesn't really address land plant evolution in relation to terrestrialization. I would suggest adding general plant evolution related to terrestrialization and linking this to C4H evolution.

Response 17: In our introduction, we described phenylpropanoid/C4H evolution in the context of plant terrestrialization (L48-67). Unfortunately, due to wording limit, we were not able to further details or introduce general aspect of plant evolution in this manuscript. We believe that our introduction conveys the necessary contextual information for the reader to fully appreciate our findings.

- The sentence from Line 215-217, "To achieve... M. polymorpha." comes across as vague and somewhat confusing. I would suggest considering its removal for clarity and coherence in the manuscript.

Response 18: Thank you for the suggestion. We have modified this sentence (L235-236).

- Regarding the phylogenetic analysis, it's noted that the ultrafast bootstrapping ('-B' option in IQ-Tree) was employed for bootstrapping, which involves pseudoreplicates. If the authors choose to reconstruct the tree with an outgroup, I recommend considering the use of the standard nonparametric bootstrapping method ('-b' option) in addition to the ultrafast bootstrapping method. However, this suggestion is optional and depends on the authors' discretion.

Response 19: Thank you for bringing this up. We are aware of alternative bootstrapping methods, which all have advantages and drawbacks. We choose to maintain the ultrafast bootstrapping which has become a standard, validated approach (PMID: 29077904; 5000+ citations).

- The key term "terrestrialization" is far too broad to be mentioned in this manuscript. I would suggest changing the key word to something more specific (e.g., CYP73 evolution, cuticle, C4H).

Response 20: Literature now plainly associates the phenylpropanoid pathway with the first steps of plants on land. We therefore believe that the key word "terrestrialization" properly contextualizes the study and complements the title. Additional, more specific key words have been added (biopolymers, cinnamic acid 4-hydroxylase) following the recommendation.

- I don't understand the Figure S2 result. Could you please explain how to interpret it in greater detail?

Response 21: Cytochromes P450 exhibit a maximum absorbance at 450 nm when bound to carbon monoxide, hence their name. This spectral property is used to determine CYP quantity and informs on the proper folding of the protein (PMID: 19661994). Misfolded/destabilized CYPs do not exhibit such absorbance maximum at 450 nm. **Appendix Fig. S2** results therefore demonstrate that R>A mutations do not significantly alter protein integrity. An explanatory sentence has been added **Appendix Fig. S2** legend.

Referee #2:

In this manuscript, the authors have undertaken a series of experiments to characterize the C4H enzyme in bryophytes, aiming to shed light on the evolution of land plants linked to the enzymatic activity of phenylpropanoid biosynthesis. They present evidence suggesting that the origin of C4H can likely be traced back to the common ancestor of land plants. Additionally, they report that C4H in algae was undetectable using their BLAST-based method, aligning with previous studies. In vitro assays were successful in confirming the conserved activity of C4H in bryophyte homologs, including the identification of a specific residue crucial for C4H activity. The study further examines knock-out phenotypes in *Physcomitrium patens* and *Marchantia polymorpha*, uncovering a conserved role in cuticle formation and highlighting its significance in development. The research is methodologically sound, albeit with few unexpected findings, and it significantly contributes to our understanding of the evolutionary conservation of the important enzyme in land plants.

The manuscript is generally well-written and provides an engaging read. However, there are specific areas where clarification or additional information would enhance the reader's understanding:

L106: The term 'homology' denotes shared ancestry. In the context of BLAST hits, the use of 'sequence similarity/identity' would be a more accurate terminology than 'homology'.

Response 22: This is a good point. We have replaced the term "homology" by "sequence identity" where relevant.

L113: The manuscript details the presence of multiple CYP73 paralogs in bryophytes but selects only one per species for functional analysis without initial justification. Although this might be explained later in the text, a brief rationale at this point would enhance reader understanding. Clarification on the selection criteria among these paralogs is recommended.

Response 23:

To ascertain that bryophyte CYP73 homologs identified in phylogenetic analysis (**Fig. 1C**) were *bona fide* C4H enzymes, we decided to investigate one candidate protein per major bryophyte groups (mosses,

liverworts and hornworts). This constituted a minimal set of proteins to conclude on the conservation of C4H activity in bryophytes. Within each group, candidate genes/proteins were selected based on genome/transcriptome data availability and quality at the time the project was initiated. For instance, in hornwort, PpCYP73A protein was selected because it had the best, full-length quality sequence scaffold in RNA-seq in the 1kP database. PpCYP73A48 and MpCYP73A1 proteins were chosen because we were concurrently investigated them *in planta*. This is now briefly explained in the text (L126-129).

L165: The description of *uidA* reporter lines is somewhat cursory. It would be beneficial to specify whether these genes are expressed in the epidermis or the inner tissues of gametophyte stems. This detail is pertinent to the discussion on 'waterproofing' capabilities introduced earlier and is also relevant to the subsequent analysis of cuticular composition and function.

Response 24: We agree that this point is relevant and therefore performed cross sections of the *CYP73:uidA* plants to address it. Histologic sections are now visible in Fig. 3E-F (see below) and show that both *CYP73A48* and *CYP73A49* are expressed in the central part/cylinder of the gametophore. *CYP73A49* in addition displays an expression in the basal part of phyllids.

Figure 3. Functional analysis of CYP73 genes in the moss *Physcomitrium patens*.

(A) Expression profiles of the four *P. patens* CYP73 paralogs in various tissues. Data are derived from the CoNekT database (<https://evorepro.sbs.ntu.edu.sg>; Proost & Mutwil, 2018). DAF, days after fertilization; TPM, transcripts per kilobase million. (B) Angiosperm and bryophyte CYP73 multiple sequence alignment centered on the protein region encompassing *t*-cinnamic acid binding residues identified by docking. At, *Arabidopsis thaliana*; Bd, *Brachypodium distachyon*; Mp, *Marchantia polymorpha*; Pc, *Phaeoceros*

carolinianus; Pp, *Physcomitrium patens*. (C) Representative UHPLC-MS/MS chromatograms of *in vitro* C4H assays performed with recombinant PpCYP73-3xHA tagged proteins. Assays performed with microsomes derived from yeasts transformed with an empty vector were used as negative control. (D) Phenotype of *Arabidopsis thaliana* three-week-old (upper panel) and six-week-old (lower panel) wild type, *cyp73a5-1* mutant and *cyp73a5-1* complemented with *AtCYP73A5*, *PpCYP73A48* and *PpCYP73A49* coding sequences. Two independent complemented lines are depicted for each gene. (E-F) Representative GUS staining pattern in two-month-old gametophores of *PpCYP73A48:uidA* (E) and *PpCYP73A49:uidA* (F) reporter lines. For each gene, the central, left and right pictures show whole gametophores, the apex of a gametophore and a gametophore cross-section, respectively. Magenta arrowheads point to phyllids. (F) Pictures of two-month-old colonies of wild type, $\Delta PpCYP73A48$ and $\Delta PpCYP73A49$ single mutants, and $\Delta PpCYP73A48/CYP73A49$ double mutant. Close-ups on individual gametophores from each genotype are visible in the lower part. (G) Pictures of five-week-old $\Delta PpCYP73A48/CYP73A49$ gametophores grown in low-melting point agarose Knop medium supplemented, or not (control), with 50 μ M *p*-coumaric acid. Magenta arrowheads point to developed phyllids.

L189: The phrase 'complete "CYP73" deficiency' seems vague. Given the existence of untargeted CYP73 paralogs, a more precise description ("C4H"?) of the deficiency would be beneficial.

Response 25: Our data point to the fact that only two out of four *CYP73* moss paralogs encode *bona fide* C4H (i.e. *CYP73A48* and *CYP73A49*), implying that corresponding double mutants are virtually completely C4H deficient. However, in order to avoid ambiguity, we have amended the text and removed the mention "complete *CYP73* deficiency" (L201).

L286: The term "evolution reconstruction" is a bit ambiguous. If it refers to 'phylogenetic analysis', using this specific term would prevent potential misinterpretation.

Response 26: Thank you for pointing toward this ambiguity. This part of the discussion has been removed following reviewer recommendation below (see **response 28**).

L338: The observation that exogenous *p*-coumaric acid did not fully rescue the phenotype suggests a considerable likelihood that the remaining unrescued phenotypes may be attributable to the hyperaccumulation of *t*-cinnamic acid.

Response 27: This is indeed a possibility we cannot rule out. However, the following observations go against this hypothesis:

1. $\Delta PpCYP73A48/CYP73A49$ double mutants, which are severely impaired in their development, do not accumulate free *t*-cinnamic acid (**Appendix Fig. S9**); they rather accumulate *t*-cinnamate esters as evidenced from the appearance of *t*-cinnamic acid in saponified extracts (**Fig. 4C**)
2. *t*-cinnamic acid level in saponified extracts do not correlate with phenotype severity since $\Delta PpCYP73A49$ mutants are far less affected than double mutants while amid similar *t*-cinnamic acid level (**Fig. 4C**).

As to the extent of chemical complementation, it is worth noting that chemical complementation of $\Delta PpCYP73A48/CYP73A49$ double mutants involves the efficient uptake of *p*-coumaric acid from the medium and its transport within the plant to the site of use (including subcellular localization). Partial achievement of either of these processes could explain an incomplete complementation.

The initial paragraph of the Discussion section could potentially be more fitting in the Introduction, as it

primarily contextualizes the research without delving into specific results.

Response 28: Thank you for the recommendation. We have discarded the first paragraph of the discussion as it was partly redundant with the introduction.

Fig. 4A x-axis label: It appears that "Time" would be a more appropriate label than "Temps".

Response 29: Well spotted, thank you! We have made the change.

Referee #3:

Knosp et al. provide new data on the functional origin of C4H enzymes in plants. Beginning with phylogenetic analysis of C4H enzymes, the authors identify and then functionally validate C4H in bryophytes through a combination of molecular genetics, biochemistry, and analytical techniques. This focused examination of phenylpropanoid pathway evolution serves as an excellent example of how to leverage non-flowering/non-vascular systems to answer key questions in plant biology and biochemistry. I am largely supportive of the manuscript overall and have only a few concerns/comments.

1. Genetic complementation/rescue: The data would be strengthened by rescuing the mutants generated in *Physcomitrium* and *Marchantia*. Moreover, the authors could provide data on the characterization of *Arabidopsis* mutant rescues (for example, westerns or RT-PCR). This is especially useful for the PpCYP line that failed to rescue *At* mutants (i.e. is this due to low transgene expression)?

Response 30: Since successful chemical complementation of moss double mutant with *p*-coumaric acid (C4H product) and *CYP73* mutant phenocopy by treatment with the C4H inhibitor piperonylic acid (PA) provide strong orthogonal evidence supporting conclusions from genetic inactivation, we did not perform genetic rescue.

We agree that the failed rescue of *Arabidopsis* mutant with the moss *CYP73A51* gene could be explained by an insufficient expression. However, the stunted growth of the *Arabidopsis cyp73a5-1* mutant (barely produces true leaves) makes it very tedious and challenging to harvest enough material for expression analysis. We therefore decided not to perform such an experiment, considering also that the absence of C4H activity of *CYP73A51* (**Fig. 3C**) and sequence variation in *t*-cinnamic acid binding sites (**Fig. 3B**) provided good evidences it was not a *bona fide* C4H.

2. Is it possible to assess cuticle integrity across all bryophytes? Given the analytical data, it could be informative to assess whether the *Marchantia* mutants are similarly impacted through toluidine blue assays. This could be further extended to the piperonylic acid treatments if feasible.

Response 31: Thank you for the suggestion, reviewer 1 also raised this point. As detailed in **Response 12**, we tried to perform toluidine blue assays in *M. polymorpha* and *A. agrestis* but faced technical difficulties related to the thalloid shape of these plants. Results could not be interpreted in reason of the massive uptake of toluidine blue by rhizoids.

3. An intriguing implication of the data is that streptophyte algae lacking C4H would be insensitive to PA treatment. Providing a wider exploration C4H inactivation via PA (algae, ferns, etc) could strengthen the evolutionary assertions mentioned throughout the manuscript. I appreciate that these systems are not trivial to set-up, so this is only a suggestion provided that the assays/experiments were feasible for the authors.

Response 32: Thank you for this very interesting suggestion. We have expanded our investigations to two additional models, the fern *Ceratopteris richardii* and the charophyte *Klebsormidium nitens*. In parallel with the three bryophyte species, these two additional species were treated with the C4H inhibitor piperonylic acid (PA). Obtained results show that PA impacts the development only in embryophytes. We did not observe any particular change in *K. nitens* upon PA treatment. These data are now available in Fig. 6 (see below) and described L269-295.

Figure 6. Developmental and metabolic effects of the C4H inhibitor piperonylic acid in streptophytes.

Unless otherwise stated, wild-type plants were treated with 50 μM piperonylic acid (PA) or corresponding mock (0.05% DMSO) on agar plates. *C. richardii* (fern) was treated for three weeks after transfer of two-week-old sporophytes. *P. patens* was treated for one month after transfer of individual gametophores. *M. polymorpha* was treated for three weeks after transfer of individual gemmae. *A. agrestis* was treated for two months after transfer of thallus pieces. *K. nitens* (charophyte, Klebsormidiophyceae) was treated in liquid cultures for two weeks after subculturing (1:10 ratio). (A) Pictures of plants after PA or mock treatment. (B) Representative UHPLC-UV chromatograms of crude metabolic extracts of mock- and PA-treated plants. *P. patens* metabolic analysis was performed on plants grown in liquid culture to facilitate tissue collection. (C) Relative quantification of shikimate phenolic esters in mock- and PA-treated plants. C5S, caffeoyl-5-O-shikimate; pC5S, *p*-coumaroyl-5-O-shikimate. Results are the mean ± SEM of three independent biological replicates for each condition. Mock versus PA unpaired *t* test adjusted *P*-value: ***P*<0.01; ****P*<0.001. nd, not detected.

4. If C4H acts early in the phenylpropanoid pathway, then how do other phenylpropanoids accumulate in *Marchantia* mutants? Is there a genetic redundancy issue here, as in *Physcomitrium*? Clarity on this could be useful, perhaps in the discussion.

Response 33: Thank you for your comment. There is indeed a *CYP73* genetic redundancy in *M. polymorpha*. We genetically inactivated the main one (*MpCYP73A1*), based on expression level, leaving the two other paralogs unaltered. We further demonstrated that the expression of these two minor paralogs

was unchanged in *Mpcyp73a1* mutant backgrounds (**Fig. 5B; L244-246**). Residual C4H activity deriving from these two intact paralogs most likely explained the fact phenylpropanoid biosynthesis is not completely abolished in *Mpcyp73a1* mutants. This is now briefly commented **L258-260**.

Dear Dr. Renault,

Thank you for submitting a revised version of your manuscript. Your study has now been seen by all original referees, who mostly find that their previous concerns have been addressed and now recommend publication of the manuscript. There is still one point raised by Referee #1 regarding your plots that we would like you to address (please see below). There remain only a few mainly editorial points that have to be addressed before we can extend formal acceptance of the manuscript:

1. Please submit up to five keywords. Currently you have more than five.
2. Please check that the funding information is correct and identical both in the manuscript and our online system. The details you provided in the Comments box in our online system need to be removed and each funder needs to be entered separately via More Funders button.
3. Please submit a complete author checklist, which you can download from our author guidelines (<https://www.embopress.org/pb-assets/embo-site/EMBO%20Press%20Author%20Checklist-1642513524327.xlsx>). Please insert information in the checklist that is also reflected in the manuscript. The completed author checklist will also be part of the Review Process File.
4. Please make sure that the order of the sections in the manuscript is as follows: abstract, introduction, results, discussion, materials & methods, data availability section, acknowledgments, disclosure statement and competing interests, references, main figure legends, tables, expanded figure legends.
5. Please make sure that your Data availability section is provided in its final form.
6. The conflict of interest section needs to be renamed to "Disclosure Statement and Competing Interests"

Figure issues:

1. FIGURE CALLOUTS: missing callouts for - Figure 3H, Appendix Tables S1-S5, Datasets EV1-EV6. These should be called out in the text.
2. DATASET EV LEGENDS: 6 zipped folders were uploaded; the legends are provided in the Appendix; they need to be removed from the Appendix file and need to be provided in each folder; They need to be renamed to Dataset EV1, etc.
- 3.
4. We can accommodate up to five EV figures. Please consider either moving two of the EV figures to the Appendix, or transforming, for example Figures EV5 and EV7 into main figures.
5. Please turn Figures S1-4 into EV figures. We replaced Supplementary Information with Expanded View (EV) Figures and Tables that are collapsible/expandable online. EV Figures should be cited as 'Figure EV1, Figure EV2' etc. in the text and their respective legends should be included in the main text after the legends of regular figures. Further information on the format is available here: <https://www.embopress.org/page/journal/14602075/authorguide#expandedview>.
6. Please remove figures from the manuscript text file, while assembling the figure legends after the "References" section.

Appendix:

7. Appendix 1 file with ToC: included, but the figures and tables are missing the word "Appendix" in their name; a ToC with page numbers is needed on a title page. Please correct.

Acknowledgements and competing interests sections:

8. CRediT has replaced the traditional author contributions section because it offers a systematic, machine-readable author contributions format that allows for more effective research assessment. Please remove the Authors Contributions from the manuscript itself and use the free text boxes beneath each contributing author's name in our online submission system to add specific details on the author's contribution. More information is available in our guide to authors.
9. Please rename "Conflict of interest" section into "Disclosure and competing interests statement" (further info: <https://www.embopress.org/page/journal/14602075/authorguide#conflictsofinterest>).

References:

10. Please make sure that the references are formatted according to The EMBO Journal style - where there are more than 10 authors on a paper, the first 10 should be listed, followed by 'et al.' Please see further information here: <https://www.embopress.org/page/journal/14602075/authorguide#referencesformat>

Synopsis:

11. Papers published in The EMBO Journal are accompanied online by a 'Synopsis' to enhance discoverability of the manuscript. It consists of A) a short (1-2 sentences) summary of the findings and their significance, B) 3-4 bullet points highlighting key results and C) a synopsis image that is 550x300-600 pixels large (width x height, jpeg or png format). You can either show a model or key data in the synopsis image. Please note that the image size is rather small and that text needs to be readable at the final size. Please send us this information together with the revised manuscript.

Please let me know if you have any questions regarding any of these points. You can use the link below to upload the revised

files.

With best regards,

Yehu Moran
Academic Editor
The EMBO Journal

General Instructions for preparing your revised manuscript:

We realize that it is difficult to revise to a specific deadline. In the interest of protecting the conceptual advance provided by the work, we recommend a revision within 3 months (12th Sep 2024). Please discuss the revision progress ahead of this time with the editor if you require more time to complete the revisions.

Referee #1:

I appreciate the author's effort in responding to many comments, despite the fact that I asked for many things. The authors addressed most of the concerns raised in the previous review and have improved their manuscript by making changes to the text and by including new data.

This is a minor comment for the authors to improve plot visibility (e.g., y-axis value) when multiple plots are combined in the z-axis (e.g., Fig 2D, Fig 3C, Fig 5F, Fig 6B), as I do not think this has been improved in the revised manuscript. Also in Fig 1D, they demonstrated that the empty vector has an MCps value of 1, whereas the standard has 0 and increases to 2.5 at the peak. I guess that the empty vector should be 0, but they plotted it on the z-axis so it appears to be in 1. I could be wrong, but this suspicion comes from the author's inconsistency in their plotting. And if the authors do not want to change their plots, it would be helpful to include at least the peak's y-value to see how the mutants compare to one another. I would recommend make all plots like Fig 4A, but this is up to the authors.

Besides that, I have no further comments and congratulate the authors on a very nice paper!

Referee #2:

I thank the authors for addressing all of my previous comments. I have no further concerns about the updated manuscript.

Referee #3:

The authors addressed my main concerns and I am pleased to see the exciting new data on piperonylic acid treatment of ferns and algae that supports the authors main claim. Happy to endorse publication.

Response to Referees

Referee #1:

I appreciate the author's effort in responding to many comments, despite the fact that I asked for many things. The authors addressed most of the concerns raised in the previous review and have improved their manuscript by making changes to the text and by including new data.

Response 01: We thank the reviewer for sharing his/her time and expertise in evaluating our work.

This is a minor comment for the authors to improve plot visibility (e.g., y-axis value) when multiple plots are combined in the z-axis (e.g., Fig 2D, Fig 3C, Fig 5F, Fig 6B), as I do not think this has been improved in the revised manuscript. Also in Fig 1D, they demonstrated that the empty vector has an MCps value of 1, whereas the standard has 0 and increases to 2.5 at the peak. I guess that the empty vector should be 0, but they plotted it on the z-axis so it appears to be in 1. I could be wrong, but this suspicion comes from the author's inconsistency in their plotting. And if the authors do not want to change their plots, it would be helpful to include at least the peak's y-value to see how the mutants compare to one another. I would recommend make all plots like Fig 4A, but this is up to the authors.

Response 02: We would like to thank the reviewer for sharing its concern regarding chromatogram displays. However, we maintain that chromatograms visible in Fig.1D, Fig. 2D, Fig 3C, Fig 5F, Fig 6B are valid and comply with accepted standards in the field (see e.g. Fig. 3A of PMID: 23950498).

With regard to Figure 1D, it should first be noted that the chromatograms are drawn to scale within the same graph, allowing direct comparisons. In this particular case, the "Empty vector" chromatogram is flat, regardless of the baseline on the Y axis, indicating that no molecules were detected. We chose to adapt Y-axis scale to each enzyme/sample as product (p-coumaric acid) levels were fairly different.

Besides that, I have no further comments and congratulate the authors on a very nice paper!

Referee #2:

I thank the authors for addressing all of my previous comments. I have no further concerns about the updated manuscript.

Response 01: We thank the reviewer for sharing his/her time and expertise in evaluating our work.

Referee #3:

The authors addressed my main concerns and I am pleased to see the exciting new data on piperonylic acid treatment of ferns and algae that supports the authors main claim. Happy to endorse publication.

Response 01: We thank the reviewer for sharing his/her time and expertise in evaluating our work.

Dear Dr. Renault,

I am pleased to inform you that your manuscript has been accepted for publication in the EMBO Journal. Congratulations on an excellent paper and an important contribution to the field of molecular plant evolution.

Yours sincerely,

Yehu Moran
Editor
The EMBO Journal
